# A diversity centric strategy for the selection of spatio-temporal training data for LSTM-based streamflow forecasting

Everett Snieder[1*] and Usman T Khan[1]

[1]Department Civil Engineering, Lassonde School of Engineering, York University, Toronto, Canada.
[*]Address correspondence to: esnieder@yorku.ca

**Abstract.** Deep learning models are increasingly being applied to streamflow forecasting problems. Their success is in part attributed to the large and hydrologically diverse datasets on which they are trained. However, common data selection methods fail to explicitly account for hydrological diversity contained within training data. In this research, clustering is used to characterise temporal and spatial diversity, in order to better understand the importance of hydrological diversity within regional training datasets. This study presents a novel, diversity-based resampling approach to creating hydrologically diverse datasets. First, the undersampling procedure is used to undersample temporal data, and is used to show how the amount of temporal data needed to train models can be halved without any loss in performance. Next, it is applied to reduce the number of basins in the training dataset. While basins cannot be omitted from training without some loss in performance, we show how hydrologically dissimilar basins are highly beneficial to model performance. This is shown empirically for Canadian basins; models trained to sets of basins separated by thousands of kilometres outperform models trained to localised clusters. We strongly recommend an approach to training data selection that encourages a broad representation of diverse hydrological processes.

## 1 Introduction

Floods constitute a major threat to populations and infrastructure and are projected to increase in severity due to climate change and urbanisation. Flood early warning systems (FEWS), which rely on models that predict streamflow, provide advanced notice of flood risk and are considered amongst the best ways to mitigate flood damage. Many Canadian communities lack any sort of FEWS, making them vulnerable to flood damage. Over the past 3 decades, machine learning (ML) models have been increasingly applied for streamflow prediction and represent significant potential for improving the accuracy and coverage of FEWS in flood prone regions. Recently, several large sample studies have shown that ML models can consistently outperform traditional, physics-based hydrological models (Mai et al., 2022; Arsenault et al., 2023a; Kratzert et al., 2019b), which underscores their proficiency for FEWS.

ML model development has typically followed the same format as physics-based models, in that a single model is parameterised and calibrated on an individual basin, which is referred to as a locally trained model. The work by Kratzert et al. (2019b) demonstrated that the accuracy of ML models can be improved by training a model to a set of basins, rather than an individual basin, which is referred to as a regionally trained model. Regional training relies on deep learning architectures such as long-short term memory networks (LSTMs), which have recently surged in popularity for streamflow forecasting applications and are considered to be state-of-the-art (Fang et al., 2022). Recent advances in regional learning have focused on improvements to model architectures (Nevo et al., 2022; Girihagama et al., 2022), and benchmarking against traditional physics-based models (Lees et al., 2021; Arsenault et al., 2023a).

Broadly speaking, physics-based and ML hydrological models benefit from diverse training data, which improves their performance on future, unseen conditions. Locally trained models are only provided with temporally diverse data at a single point in space. In contrast, regionally trained models have been empirically shown to outperform locally trained models in several works (Kratzert et al., 2019b; Zhang et al., 2022), and their success is in part attributable to the spatial and temporal hydrological diversity contained in the multi-basin datasets on which they are trained (Kratzert et al., 2019b).

However, there is currently little guidance on optimum basin (spatial) selection of training data for these models. Often times, models are trained to complete large-sample datasets such as those included in Caravan (Kratzert et al., 2023). However, with increasingly large, global hydrological datasets, it is not always practical or feasible to train to all available data, especially when conducting computationally expensive tasks such as hyperparameter selection, which is required to achieve optimum model performance, or creating multimodel ensembles. Therefore, there is a need for improved guidance on efficient methods for training data selection, to maximise model performance and generalisation. Many of the deep learning advances in hydrology (e.g., (Kratzert et al., 2019a), (Klotz et al., 2022), Lees et al. (2022), and (Gauch et al., 2021a)) have utilised well-established large-sample basins (see Caravan (Kratzert et al., 2023) and the datasets therein). Canada is a country with a diverse hydrological landscape, characterised by coastal regions, mountain, urban, and exposed rock in the form of the Canadian shield. The effectiveness of regionally trained models has not yet been well established on highly diverse Canadian basins.

When training a model to predict streamflows in some region of interest, the goal is to select the most relevant spatial and temporal data, while avoiding data that has either no impact or a negative impact on model performance. Unsupervised clustering has been used in previous studies as a data-driven approach to identify spatial and temporal diversity within a training dataset (Toth, 2009; Kratzert et al., 2024). The application of clustering as a means to identify spatial and temporal diversity is in itself nothing new. Many studies, which are reviewed below, have applied clustering to spatial and temporal data as a means to quantify hydrological diversity. However, the treatment of hydrological diversity generally follows one of two approaches: either, it is used to generate hydrologically diverse datasets, or datasets with homogeneous hydrological conditions. The former, aims to generalise models to a wide range of conditions, promoting balanced performance and good generalisation, while the latter aims to simplify the learning problem, improving performance in similar conditions. Both approaches have been used successfully for temporal streamflow clustering. Anctil and Lauzon (2004) apply a self-organising map (SOM) to streamflow data in a single basin to create a training dataset with a balanced representation of diverse hydrological states. In contrast, Toth (2009) use an SOM to classify streamflow into homogeneous subsets, on which individual models are trained and combined in a modular format. Their approach is found to improve overall prediction accuracy, which can be attributed to the error diversity of the collection of trained models. Snieder et al. (2021) partition streamflows into typical streamflows and high streamflows, in order to undersample typical streamflows and oversample high streamflows, which is found to improve performance on the latter, which is desirable for FEWS applications. The same motivation has led to numerous applications of clustering on basins, particularly in regional training schemes. For example, Gauch et al. (2021b) showed that implicitly increasing hydrological diversity of regional training datasets, by iteratively increasing the number of basins, as well as the amount of data in each basin, improves model generalisation. However, the study does not explicitly quantify hydrological diversity, in part due to the absence of a widely agreed upon metric for hydrological similarity (Oudin et al., 2010). However, other studies have used clustering to estimate hydrological diversity, such that basin selection can explicitly account for hydrological diversity. These cases tend to use some form of clustering (either supervised or unsupervised) to quantify hydrological diversity within training data and the effects it has on model generalisation. Zhang et al. (2022) applied K-means clustering to a set of 35 mountainous basins in China based on hydroclimatic attributes, finding that a model trained to all available basins typically outperformed those trained to individual clusters. Hashemi et al. (2022) applied a similar approach by classifying basins into

distinct hydrological regimes based hydrometeorological thresholds. As done in Zhang et al. (2022), their study compared locally and globally trained models, finding only minor differences in the performance between the two. A common problem in comparing global and locally trained models is that these comparisons typically do not control for sample size. As a result, the improved performance of the global model can be impacted by the regularisation effect on the sample size. In other words, deep learning models trained to small datasets may be overfitted and thus, poorly generalised. Fang et al. (2022) accounts for this potential issue. Their study used existing 'ecoregion' basin classification, which were classified by the United States Environmental Protection Agency, and evaluates the effects of additional training basins at three similarity intervals. Their study showed that counterintuitively, 'far' or 'dissimilar' basins amongst the training dataset often produced greater improvements in model performance when compared to the inclusion of 'close' basins. They speculate that distant basins provide a regularisation effect. Fang et al. (2022) call for further investigation into the effect that hydrological diversity has on model generalisation and underline the need for a systematic approach. Kratzert et al. (2024) characterise hydrological diversity by applying K-Means clustering to basin attributes, finding that models trained to basins with similar hydrological characteristics outperform randomly sampled basin sets of the same sizes. However, in every case, they show that LSTMs trained to hundreds of basins outperform those trained to smaller subsets. They also demonstrate how regional learning improves performance on extreme events, thus for FEWS, as the training datasets contain a higher number of extreme events, spread across all basins. Many of these studies assume that similar basins are most useful to one another in the context of regional learning. We challenge this assumption, and seek to determine to what extent hydrologically similar data is beneficial for training.

The objective of this study is to study the effect that the formation of hydrologically diverse training datasets has on model performance and generalisation. Hydrological diversity is quantified using clustering, which is applied separately to streamflow (temporal) and basins (spatial). This topic is analysed throughout two experiments. In the first experiment, we evaluate the effects of removing non-diverse streamflow data from regional training datasets. The latter test is repeated, but by undersampling non-diverse basins, instead of streamflow values, from a larger subset. In the second experiment, we compare the effects of adding 'similar' and 'dissimilar' basins to a training dataset for some region of interest. The purpose of this second experiment is to compare the contribution of additional basins to model generalisation, with respect to their hydrological similarity to the evaluation set.

While numerous studies have applied clustering to streamflow and basins, to the extent of the knowledge of the authors, the use of clustering to explicitly create spatial and temporally diverse training datasets is a novel approach. The outcome of these experiments has the potential to improve methods for the creating of training datasets for regionally trained models. This topic is investigated on sequence-to-sequence (Seq2Seq) LSTMs for daily streamflow forecasts of 1–3 days in Canadian basins. Basins are sampled from across Canada and models are trained using historic hydrometeorological data from the past 36 years.

## 2    Methods and Data

### 2.1    Input and target variables

This study uses data retrieved from the HYSETS dataset, which contains hydrometeorological data for over 14,000 basins across North America (Arsenault et al., 2020). The target variable, the future state of streamflow, is predicted using dynamic and static input features. The models in this study are autoregressive (AR), meaning past streamflow at the target gauge is used as one of the input features (Nearing et al., 2022). Additional dynamic features from the HYSETS database includes daily basin-averaged minimum temperature, maximum temperature, precipitation, and snow water equivalent (SWE), which are listed in Table 2. AR LSTMs are used since they are more accurate than non-AR LSTMs (Nearing et al., 2022), and because the Canadian hydrometric network is largely available in real-time. The static basin attributes, which are summarised in Table 1 in the appendix, allow the model to transfer learned information between basins. Some of the static attributes are included in the HYSETS database, while additional attributes are calculated based on the dynamic timeseries, which generally follow those included in the CAMELS dataset (Addor et al., 2017). The additional attributes are calculated based on the timeseries' data already contained within HYSETS and do not rely on any external databases. Unfortunately, there are no attributes that reveal whether a catchment is regulated by built infrastructure. However, the model may still be able to learn the effects of built infrastructure implicitly, through the rainfall-runoff relationship. Lastly, the input feature set also contains a one-hot encoded basin label (Lees et al., 2022), which enables the model to distinguish between streamflows in different basins. Static basin attributes are used both as input features for the streamflow forecasting model and in the basin clustering method, which are described in Sections 2.3 and 2.3, respectively.

### 2.2    Basin selection

This study only considers Canadian basins from the HYSETS database. Basin are removed if they have less than 80% data availability within any of the training, validation, and testing periods, which span a total of 36 years from Oct. to Sep. of 1982-1994, 1994-2006 and 2006-2018, respectively, following the split-sample method (KLEMEŠ, 1986). The training partition is used to train the models, validation for fine-tuning LSTM hyperparameters, and the test partition is used to calculate model performance. While records in some basins exist prior to 1982, it is imperative that the data used to train and evaluate basins be from the same time period and be of a similar size. Including records from before 1982 results in fewer basins to choose from, which tends to reduce the hydrological variability of the basin set. Next, some basins are removed, due to missing static attributes. These criteria produce a set of approximately 2000 basins, with highly variable attributes, according to Table 1.

### 2.3    Sequence-to-sequence LSTM models

LSTM models, with a Seq2Seq architecture (Cho et al., 2014), are used to generate forecasts at a daily resolution, at multiple lead times. Seq2Seq models are composed of an encoder and decoder; the encoder transforms an input sequence into a fixed length context vector, which is provided to the decoder, which outputs predictions. Recently, several studies have demonstrated

the aptness of Seq2Seq models for predicting runoff at multiple lead times (Xiang et al., 2020; Girihagama et al., 2022; Zhang et al., 2022).

    The Seq2Seq models in this study use hyperparameter values that are common for LSTM rainfall runoff models. The models in this study use a hidden layer size of 128 cells, a dropout rate of 0.2, a batch size of 32, and Adam optimisation with a decaying learning rate of 0.001 to 0.0001 across a total of 80 epochs. Input and output sequences of 7 and 3 days were used, respectively.

While an input sequence of 365 days is commonly used for streamflow prediction (Kratzert et al., 2019b; Arsenault et al., 2023b). Gauch et al. (2021b) noted that small sequences are better suited to small basin sets, and have been used in AR models (Nevo et al., 2022).

    Typically, for regional training, the error terms of individual basins are normalised based on streamflow variance of that basin. Using the typical variance-based regularisation applied to the cost function produces a relative increase in weight applied to

low variance basins. While this works well for non-AR models, AR models have a tendency to develop an over-reliance on recent streamflow observations, which can manifest in a positive timing error (Snieder, 2019). The resulting models may be barely distinguishable from the naive model (i.e., the most recent streamflow observation). Highly seasonal, naturalised basins are most prone to this problem, as streamflow tends to change gradually with time, thus a model that outputs recent streamflow observations might be mistakenly seen as accurate. Specialised performance metrics such as the Persistence Index (PI) are

often used to identify this problem in real-time forecasting applications (Nevo et al., 2022). This is simply because the PI normalises error relative to the naive model, with a PI less than 0 corresponding to a non-informative forecast relative to real-time observations. The same is not the case for widely used Nash Sutcliffe Efficiency (NSE), which can be misleadingly high in the same cases (Knoben et al., 2019). For this reason, we propose that basin persistence (i.e., mean squared deviation between observations at the current time and forecast time) be used to regularise the cost function for regionally trained models. Instead

of placing more weight on low variance basins, persistence-based regularisation places more weight on basins that have low error between recent and future streamflow values. Failure to do so results in models that are not adequately trained in those basins, and produce non-informative forecasts (relative to the naive model).

    In this study, basins are normalised using the formulation proposed in (Kratzert et al., 2019b) for NSE*, but substituting the basin variance for the persistence corresponding to the forecast lead time. The persistence-based cost function PI* is given by:

$$\text{PI}^* = \frac{1}{B} \sum_{b=1}^{B} \sum_{t=1}^{T} \frac{\sum (q_t - \hat{q}_t)^2}{\sum (p_b + \epsilon)^2} \tag{1}$$

    in which $q_t$ is the observed streamflow, $\hat{q}_t$ is the predicted streamflow, $\epsilon$ is a constant (0.1) that prevents the function from exploding to negative infinity (Kratzert et al., 2019b), and $p_b$ corresponds to the persistence of an individual basin $b$ in a set $B$ basins, given by:

$$p_b = \sqrt{\sum_{t=1}^{T}(q_t - q_{(t-L)})^2} \qquad (2)$$

in which $L$ is the forecast lead time of $\hat{q}_t$.

## 2.4   Performance metrics

Models are evaluated using two performance metrics: NSE and PI. The NSE, given in Eqn. 3, is amongst the most widely used metrics for hydrological models and effectively normalises the mean squared model error based on streamflow variance. The PI, which is given in Eqn. 4 and used in the basin regularisation function described above, is a similar metric but instead of

normalising squared residuals using the mean, it normalises forecasts based on the squared error between the streamflow at the current and forecast timesteps (Kitanidis and Bras, 1980).

$$\text{NSE} = 1 - \frac{\sum_{t=1}^{T}(q_t - \hat{q}_t)^2}{\sum_{t=1}^{T}(q_t - \bar{q})^2} \qquad (3)$$

$$\text{PI} = 1 - \frac{\sum_{t=1}^{T}(q_t - \hat{q}_t)^2}{\sum_{t=1}^{T}(q_t - q_{t-L})^2} \qquad (4)$$

     where $\hat{q}_t$ to the predicted streamflow, and $\bar{q}$ to the mean observed streamflow. $q_{t-L}$ is the observed streamflow, shifted by the

lead time $L$ of the forecast such that it represents the real-time observable streamflow in an operational context. Both metrics range between $-\infty$ and 1, with 1 being perfect and values less than 0 indicating performance worse than each respective baseline.

## 2.5   Clustering

Clustering is a simple yet effective way to identify hydrologically diverse data for training streamflow forecasting models. This

study uses clustering to identify two forms of hydrological diversity. First, it is applied to streamflow records of individual basins, to identify diverse streamflow conditions. Second, it is applied to static basin attributes, to identify basins with diverse hydrometeorological attributes. Note that both methods are independent of one another. This study uses the constrained K-means clustering algorithm (Bennett et al., 2000), which allows for the specification of a minimum cluster size. This avoids a problem that occurs with clustering streamflow, which is that infrequent flood streamflows typically produce a very small

cluster, constraining the number of samples that are available when drawing an even number of samples from each cluster (Toth, 2009).

The first application of clustering is to identify hydrologically diverse streamflows. Previous studies have applied clustering to the input vectors of ML models (Anctil and Lauzon, 2004; Abrahart and See, 2000). However, such approaches do not guarantee that streamflow is the main variable by which clusters are discriminated. For that reason, we engineer a feature set for clustering streamflow based solely on the target streamflow data, which encourages diverse streamflow conditions between clusters. The engineered feature set includes streamflow ($q_t$), two streamflow gradient features (given as $(q_{t-3} - q_t)/3$ and $q_{t-1} - q_t$), and two day of year features (given as $\sin(2\pi t/365)$ and $\cos(2\pi t/365)$ where t is the day of year 1,...,365. While the day of year is discontinuous between 365 and 1, the sin and cos decomposition offers a continuously changing pairing across each year, which increases the likelihood of clusters spanning from December to January. The streamflow gradient encourages the representation of rising and falling limbs within the clusters, which are not distinguishable using only the streamflow state.

The second application is on basins, which are clustered based on their static attributes. Due to the large number of features (39) and collinearity between features, principal component analysis (PCA) is used to reduce the feature set to 8 principal components. Names and statistics of the static attribute set are provided in Table S1 in the supplementary information.

Both clustering applications are used to inform a simple resampling procedure that aims to maximise hydrological diversity. The cluster-based undersampling (CUS) procedure is as follows. Given $N$ training examples (either streamflow samples or basins):

1. Select an undersampling rate $\phi$ as a fraction of $N$ and a number of clusters $K$

2. Cluster records into $K$ clusters with a minimum cluster size of $\phi N/K$

3. Sort samples based on distance to the cluster centroid

4. Select samples 1 to $\phi N/K$ from each cluster to form the training dataset

Each undersampling strategy is illustrated in Fig. 1. Subplots (a), (c), and (e) illustrate the raw streamflow timeseries (a), clustered streamflow (c), and undersampled streamflow (f). Note how the timeseries in (f) contains fewer typical streamflows and proportionally more high streamflows, compared to the continuous record in (a) and (c). Similarly, subplots (b), (d), and (f) show Canadian basins (a), clustered basins (d), and diverse, undersampled basins (f).

CUS applied to streamflow and basins are denoted as $\text{CUS}_\text{Q}$ and $\text{CUS}_\text{B}$, respectively. Selecting an equal number of examples (streamflow or basins) from each cluster results in a balanced variety of hydrological conditions within the training set. There are several reasons why such a training dataset is desirable. First, balanced hydrological conditions encourage balanced performance across different streamflows, or basins. Models trained to imbalanced datasets, such as streamflow records in which low streamflows drastically outnumber high streamflows, may be biased towards low streamflow conditions (Snieder et al. (2021)). The same reasoning applies to basin selection. A regionally trained model may be biased towards areas with dense spatial coverage. By selecting an equal number of each 'type' of basin, we encourage balanced spread of hydrological characteristics in the training basin dataset, which translates to good generalisation across a broader range of basins.

Due to the large and diverse feature set, feature importance is calculated to interpret the dominant basin attributes that distinguish clusters. Since K-means does not inherently quantify feature importance, a random forest (RF) classifier is used

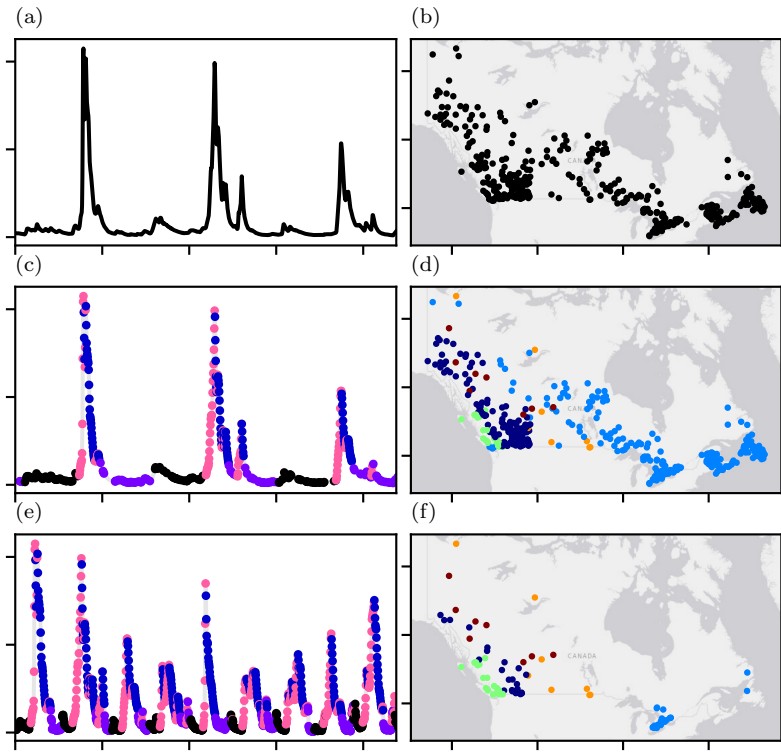

**Figure 1.** Left column, from top to bottom: unclustered (a), clustered (c), and undersampled streamflow (e) for basin 01AD003 (HYDAT ID; located along St. Francis River in New Brunswick). Right column, from top to bottom: unclustered (b), clustered (d), and undersampled basins (f). Cluster colours are arbitrary and there is no connection between temporal (a, c, e), and spatial (b, d, f) cluster colours. World Gray Canvas basemap in (b, d, f) provided by ESRI.

as a surrogate for approximating feature importance. RFs contain an intrinsic importance metric that is commonly used in hydrology (Tyralis et al., 2019). An RF with 256 estimators, and a max depth of 6 is used. Since the RF is simply fitting the outcome of the constrained K-means clustering, the RF is expected to achieve near perfect accuracy, without the need for hyperparameter tuning.

Note that the two clustering applications described above are standalone and can be combined by using them in series. While
a unified, spatiotemporal clustering method could be used to cluster samples in a single step, separating spatial and temporal clustering allows for each method to be assessed independently of one another.

**Table 1.** Streamflow resampling configurations in Experiment 1a. Labels denote the resampling type, number of clusters (K; for CUS), and resampling rage ($\phi$).

| label (K, $\phi$) | unique basins | samples (per basin) | total samples (thousands) |
|---|---|---|---|
| baseline | 64 | 4380 | 280 |
| $\mathrm{CUS_Q}$ (K=6, $\phi$=0.25) | 64 | 1095 | 70 |
| $\mathrm{CUS_Q}$ (K=6, $\phi$=0.50) | 64 | 2190 | 140 |
| $\mathrm{CUS_Q}$ (K=12, $\phi$=0.50) | 64 | 2190 | 140 |
| $\mathrm{RUS_Q}$ ($\phi$=0.50) | 64 | 2190 | 140 |

## 2.6 Experiments

### 2.6.1 Experiment 1: evaluating streamflow and basin redundancy in training datasets

In this experiment, CUS is applied to streamflow (experiment 1a) and separately applied to basins (1b). These experiments are designed to determine the extent to which non-diverse (i.e., redundant) data can be removed from training datasets, without any loss in model performance. The CUS generated datasets are compared with random undersampled (RUS) datasets, which consist of $\phi$N samples (sampled without replacement).

To evaluate $\mathrm{CUS_Q}$ (experiment 1a), a set of 64 randomly sampled basins is established. $\mathrm{CUS_Q}$ is applied to each basin individually, then merged to form the training dataset. Several resampling configurations are considered, including $\phi$ values of 0.25 to 0.50, and $K$ values 6 and 12. These configurations are compared against two baseline models, which are trained to (1) the entire dataset and (2) an RUS, which are also undersampled at 0.25 and 0.50 to match the sample size of the $\mathrm{CUS_Q}$ datasets. The parameters for each training configuration are listed in Table 1.

The framework outlined above is replicated to evaluate the effects of spatial undersampling (experiment 1b). Beginning this time with a set of 128 randomly sampled basins, subsets of basins sampled for varying numbers of clusters. Basin subsets comparisons are made against a baseline model that is trained to the entire set of basins, and RUS subsets. The purpose of this experiment is to determine whether clustering can effectively be used to identify a subset of basins that is sufficiently hydrologically diverse such that it can be used to train a model capable of generalisability on the complete set. In this experiment, the basins are trained to the entire streamflow records (i.e., no temporal resampling). The parameters for each training configuration are listed in Table 2.

### 2.6.2 Experiment 2: cross-comparison of 2 clusters of basins

The next experiment is designed to determine to what extent hydrologically dissimilar basins are useful to one another for model training. In experiment 2a, basins are divided into two clusters (which are referred to as C0 and C1), using the K-means method described in Sec. 2.5. The reasoning behind 2 clusters is to maximise the hydrological dissimilarity between basins in each cluster (based on the static basin attributes). In the first experiment, for each cluster, a baseline model is trained to 32 basins that belong to that cluster. Next, we compare the effects of adding 32 similar basins (labelled as '+similar'), with adding

**Table 2.** Basin resampling configurations in Experiment 1b. Labels denote the resampling type, number of clusters (K; for $CUS_B$), and resampling rage ($\phi$).

| label<br>(K, $\phi$) | unique<br>basins | samples<br>(per basin) | total samples<br>(thousands) |
|---|---|---|---|
| baseline | 128 | 4380 | 560 |
| $CUS_B$ (K=2, $\phi$=0.50) | 64 | 4380 | 280 |
| $CUS_B$ (K=8, $\phi$=0.50) | 64 | 4380 | 280 |
| $CUS_B$ (K=32, $\phi$=0.50) | 96 | 4380 | 420 |
| $RUS_B$ ($\phi$=0.50) | 64 | 4380 | 280 |
| $RUS_B$ ($\phi$=0.75) | 96 | 4380 | 420 |

32 dissimilar basins to the training set (i.e., from the other cluster; labelled as '+dissimilar'). In all cases, only the original 32 basins are evaluated (those used to train the baseline model); the performance of the additional training basins is not reported. This produces five unique training sets: 32 basins in cluster 0, 64 basins in cluster 0, 32 basins in cluster 1, 64 basins in cluster 1, and 32 basins in each cluster 0 and 1.

Next in experiment 2b, experiment 2a is repeated, but with cluster-based streamflow undersampling applied to the training dataset. The resulting models are trained to half the amount of training data as those in 2a. This experiment provides a comparison point between models trained in 32 basins without $CUS_Q$ and models trained on 64 basins with $CUS_Q$, as both configurations have the same number of samples. The 32 basin configurations have greater temporal representation within the evaluation basins, while the 64 basin $CUS_B$ configurations have greater spatial diversity, at the expense of temporal data. These comparisons reveal which is more useful to model generalisation: temporal data from within the subject basin, or data from outside the basin. One distinction in this experimental configuration is that it does not use one-hot-encoded basin labels, as many of the test basins are not included in the training dataset. The labels are removed to ensure that the model does not develop any dependencies on them, since they are not available for the test basins.

**Table 3.** LSTM size and training datasets used in Experiment 2. Configurations are grouped by experiment

| exp. | code | cells | n basins<br>(k=0) | n basins<br>(k=1) | training data<br>(years) | n samples<br>(thousands) |
|---|---|---|---|---|---|---|
| exp. 2a | cluster(0)-n(32) | 128 | 32 | 0 | 12 | 140 |
| exp. 2a | cluster(0)-n(64) | 128 | 64 | 0 | 12 | 280 |
| exp. 2a | cluster(1)-n(32) | 128 | 0 | 32 | 12 | 140 |
| exp. 2a | cluster(1)-n(64) | 128 | 0 | 64 | 12 | 280 |
| exp. 2a | cluster(0,1)-n(64) | 128 | 32 | 32 | 12 | 280 |
| exp. 2b | cluster(0)-n(32) cus | 128 | 32 | 0 | 6 | 70 |
| exp. 2b | cluster(0)-n(64) cus | 128 | 64 | 0 | 6 | 140 |
| exp. 2b | cluster(1)-n(32) cus | 128 | 0 | 32 | 6 | 70 |
| exp. 2b | cluster(1)-n(64) cus | 128 | 0 | 64 | 6 | 140 |
| exp. 2b | cluster(0,1)-n(64) cus | 128 | 32 | 32 | 6 | 140 |

# 3 Results and discussion

## 3.1 Experiment 1a: cluster-based temporal undersampling

Examples of temporal clustering results are provided in Figs. 2 for basin 01AD003 and 3 for basins 01AD003 and 07AF002 (located along the McLoed River in Alberta), which each belong to different basin clusters from Sec. 3.3. These results are for 6 clusters and minimum cluster sizes of 365. Although associations between clusters and specific hydrological characteristics can be expected to vary between individual basins, the results from basins 01AD003 and 07AF002 characterise four seasonal periods, as well as rising and receding limbs. Distinguishing between rising and falling limbs is consistent with previous studies that used streamflow clustering (Toth, 2009). Ensuring that distinct seasons are represented in the clustering results is important, as streamflow drivers are known to change throughout the year.

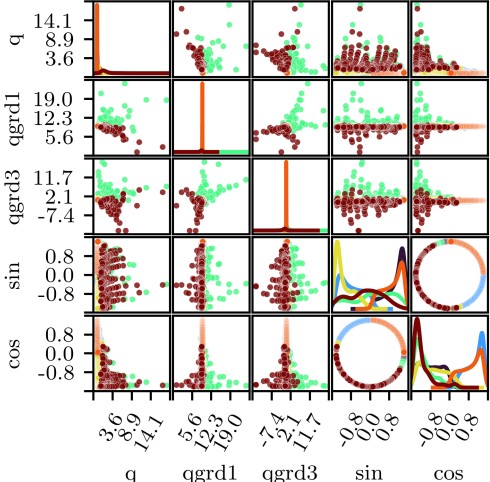

**Figure 2.** Scatterplot matrix illustrating the temporal clustering results for Basin 01AD003 streamflows (K=6). Markers are colourised by cluster and kernel smoothed histograms are shown along the diagonal. Axis labels q, qgrd1, qgrd3, sin, and cos, are shortform for $q_t$, $(q_{t-3} - q_t)/3$, $q_{t-1} - q_t$, $\sin^{-1}(t/365)$, and $\cos^{-1}(t/365)$, respectively.

The performance of models trained on a set of 64 randomly sampled basins is shown in Fig. 4 in terms of NSE (a-c) and PI (d-f) for three cases: without resampling, with cluster-based temporal undersampling ($\text{CUS}_Q$), and random temporal undersampling. The cumulative density functions (CDFs), which have an optimum shape '⌐', represent the proportion of basins (in the evaluation set) that fall below the performance along the x-axis. The baseline model (no resampling) performs reasonably well across basins, with 100% and 75% of basins achieving an NSE greater than 0.5 at the 1-day and 3-day lead times, respectively. Roughly 95% of basins achieve a positive PI, indicating lower error than the naive model.

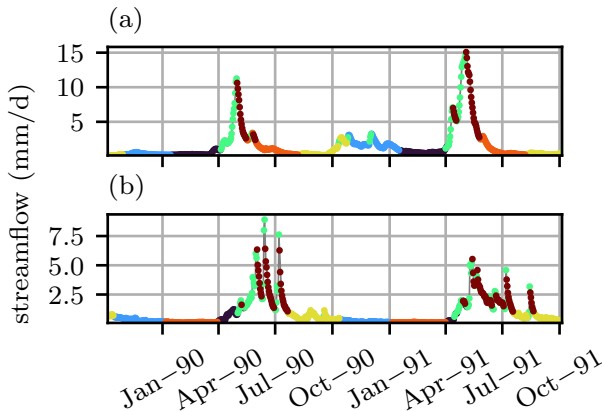

**Figure 3.** Hydrographs for Basins 01AD003 (top) and 07AF002 (bottom) with observations from October 1989–October 1991, colourised by cluster. Cluster colours are arbitrarily assigned.

In comparison, the basin sets with $CUS_Q$ at a rate of 0.5, (meaning that they use 6 out of 12 years of available training data),
achieve the same level of performance as the baseline, for both the 6-cluster and 12-cluster cases. The RUS model trained to
6 years of randomly sampled data performs very poorly. Similarly, the $CUS_Q$ model trained to 3 years of cluster-based data
perform poorly, indicating that key hydrological processes are no longer sufficiently represented in the reduced training data.
This also indicates that a lower limit of the extent to which $CUS_Q$ can be used is somewhere between an undersampling rate
of 0.25 and 0.5. Finally, in no cases do any undersampled configurations outperform the model trained to all data.

These results highlight how a simple clustering method can be used to efficiently identify subsets of data that are sufficiently
representative of the hydrological processes, which are identified by the clustering method, contained in each basin such
that there is no loss in temporal generalisation. In addition, these results show that a significant proportion of hydrological data
within a continuous series is redundant and needlessly adds to the computational burden of training, which is especially relevant
to computationally expensive tasks such as hyperparameter optimisation (HPO). Reducing the computational requirement of
HPO speeds up model development, or allows for more extensive HPO, potentially improving model accuracy, thus FEWS
reliability. Improvements in runtimes, which are reported in Tables S3 and S4, were typically found to be proportional to
undersampling rates.

A limitation of this experiment, and the subsequent experiments, is that they were conducted using AR LSTMs. If AR
inputs are not available, the model hyperparameters would need to be reconfigured and model performance would be expected
to decrease. While the experimental results are expected to transfer to non-AR models, this would need to be empirically
confirmed.

they would need to be repeated to confirm transferability.

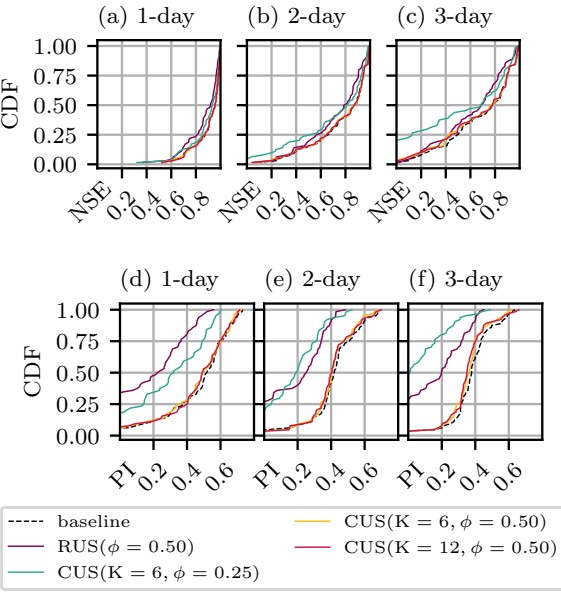

**Figure 4.** CDFs showing model performance according to NSE (top row) and PI (bottom row) for RUS (red), two $CUS_Q$ configurations (blue and green) and no resampling baseline (black). Subplots (a), (b), (c), and (d), (e), (f) correspond to forecasts of 1, 2, and 3 days respectively.

## 3.2 Experiment 1b: cluster-based spatial undersampling

In experiment 1b, $CUS_B$ is is used to select training basins. As with temporal undersampling, reducing the number of training basins required to train models has the potential to drastically reduce the computational demand of training, especially across large domains such as the thousands of gauged basins spread across Canada.

First, a baseline model is trained to a set of 128 randomly sampled basins. The basins are then grouped into K clusters, sampled at rates of 0.5 and 0.75. As with the previous experiment, RUS configurations are included at the same resampling rates, which in this case, consists of models trained to randomly sampled basin sets. The CDFs for each training configuration are shown in Fig. 5. Unlike with the temporal streamflow undersampling in experiment 1a, the basin subsets are unable to match the performance of the baseline model that is trained to the complete set of basins, which is visibly shown by the CDFs for PI. As expected, the configurations with a greater number of training basins perform most closely to the baseline. The $CUS_B$ configurations narrowly outperform the RUS configurations with the same undersampling rates, most of all at a rate of 0.50.

## 3.3 Experiment 2: cross-comparison of 2 clusters of basins

To better understand the extent to which including hydrologically similar basins in the training dataset can benefit model performance, we consider an extreme case in which basins are grouped into 2 clusters, which are referred to as C0 and C1. The choice of 2 clusters is based on the maximum silhouette score, which is a commonly used measure of cluster cohesion

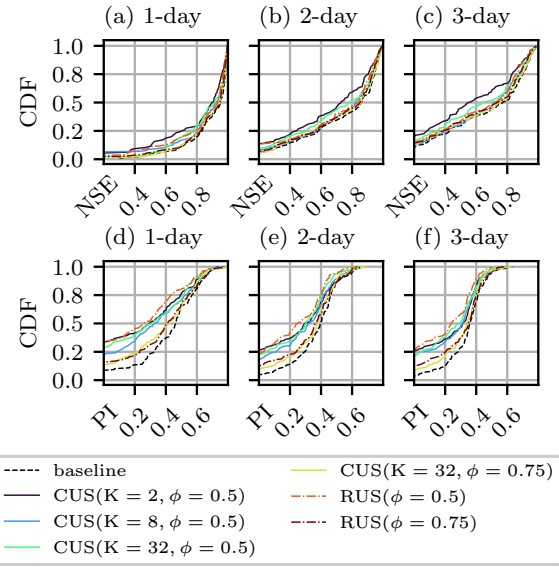

**Figure 5.** CDFs for NSE (top row) and PI (bottom row) for models trained to various basin subsets, which include a baseline (black line, includes all basins), cluster-based undersampling (coloured solid lines), and random undersampling (coloured dashed lines). The number of basins sampled from each cluster in the $\mathrm{CUS_B}$ configurations is equal to B/K. Subplots (a), (b), (c), and (d), (e), (f) correspond to forecasts of 1, 2, and 3 days respectively.

(Rousseeuw, 1987). An important note on the result of the maximum silhouette score is that the result depends on the use of constrained K-means clustering, which uses a minimum cluster size of 64. A lower minimum cluster size tends to increase the optimum number of clusters, details are provided in the supplementary information. Another reason for using two clusters is that it simplifies the analysis. For example, we compare the effects of similar, and dissimilar clusters; in contrast, a greater number of clusters would require a larger number of training configuration at varying degrees of hydrological similarity

Fig. 6 shows the spatial distribution of the cluster labels across Canada. C0 basins tend to be located at low elevations, along coastlines and east of the Rocky Mountain range. In contrast, the C1 basins are mainly confined to higher elevations in the Rocky Mountains. Mean basin attributes for each cluster are provided in Table S2 in the supplementary information.

The five most important features, identified by applying an RF to unsupervised clustering outcome, include elevation, slope, and landcovers, are shown in Fig. 7. The relevant features identified here by unsupervised clustering are consistent with the relevant descriptors deemed significant for determining hydrological similarity in the model-based method referenced in (Oudin et al., 2010).

First, for each cluster, a model is trained to a set of 32 basins from that cluster. To measure the value of adding hydrologically similar basins, 32 additional basins are added to the baseline training dataset. Finally, to measure the effects of dissimilar basins, 32 dissimilar basins are instead added to the baseline training set. Performance metrics are only calculated for the baseline set of 32 basins. The above is repeated for each cluster.

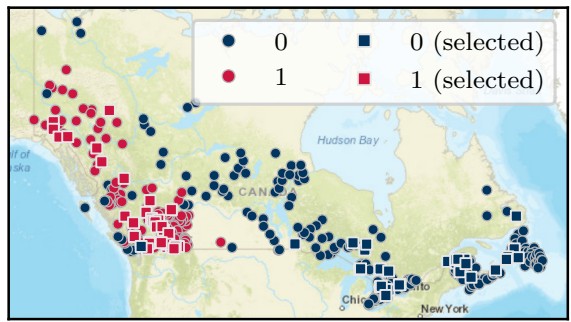

**Figure 6.** Clustering (K=2) result for basins across Canada. For each cluster, 32 square markers indicate basins selected for the baseline and evaluation sets in the two-cluster experiments. World Street Map basemap provided by ESRI.

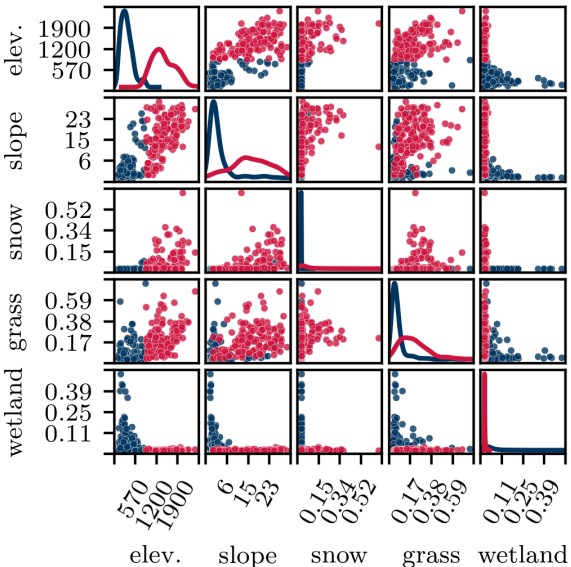

**Figure 7.** Scatterplot matrix illustrating the five most important features for clustering, as determined using the RF surrogate model. Markers are colourised by cluster and kernel smoothed histograms are shown along the diagonal.

Figs. 8 and 9 show the performance of the models evaluated on C0 and C1 basins, respectively. In C0 basins, more similar basins produce a notable improvement in performance across all lead times. Adding dissimilar basins instead produces even greater improvements, most notably, according to the PI. The same trends are seen with the C1 forecasts, with an even greater difference between the scores of the '+ similar' and '+ dissimilar' training sets.

In C0 basins, adding more similar, or dissimilar basins both improve the performance, with dissimilar basins producing larger improvements across all lead times and both metrics. A similar result is observed with the C1 basins, with the addition of dissimilar basins causing comparatively greater improvements in performance.

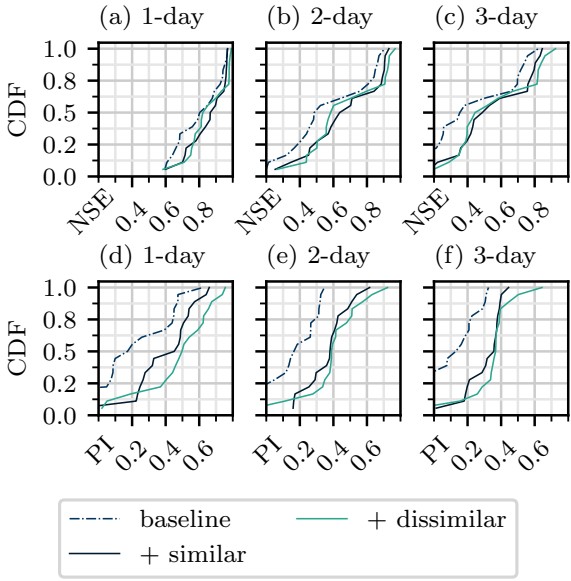

**Figure 8.** CDFs for models evaluated on C0 basins according to NSE (top row) PI (bottom row). Each row compares models trained to three different training datasets: a baseline that includes 32 basins in the respective cluster, the baseline plus 32 basins in the same cluster, and the baseline plus 32 basins in the other cluster. Subplots (a), (b), (c), and (d), (e), (f) correspond to forecasts of 1, 2, and 3 days respectively.

In experiment 2b the configurations from 2a are repeated, but incorporating $CUS_Q$, which is introduced in Sec. 3.1. Models from 2a and 2b are compared in Fig. 10 for C0 and C1 evaluation sets. Consistent with the results from experiment 1a in Sec. 3.1, $CUS_Q$ is found to have very little impact on model test performance, which is illustrated by the similar results shown when comparing results without and with $CUS_Q$ undersampling, in subplot pairs (a)-(c), (b)-(d), (e)-(g), and (f)-(h). This result indicates that for fixed training dataset size, data from outside the region of interest is much more useful to the training procedure than redundant data within the region of interest, identifiable using the clustering procedure.

### 3.3.1 Discussion

Collectively, these results reveal that the addition of training basins with distinctive hydrological characteristics is more useful in terms of improving model performance when compared to the addition of basins with similar characteristics. This outcome might be counter-intuitive, since one could expect that training to additional basins that are most similar to the test set would be the most useful. This result also highlights the danger of training models to a hydrologically similar basin sets, which is a common approach in literature (Kratzert et al., 2024; Hashemi et al., 2022).

One explanation for this result is that the input feature set is missing key explanatory variables. Two basins could have similar input vectors but different corresponding streamflow values. This difference may be explained by processes that are not captured within the input features. For example, two basins may appear to be hydrologically similar based on the available basin attributes, may have a different rainfall-runoff relationship, due to factors not included in the basin attributes, such as

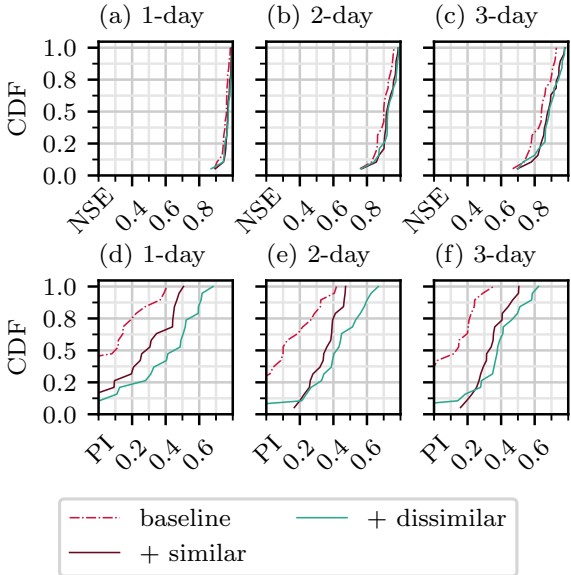

**Figure 9.** CDFs for models evaluated on C1 basins according to NSE (top row) PI (bottom row). Each row compares models trained to three different training datasets: a baseline that includes 32 basins in the respective cluster, the baseline plus 32 basins in the same cluster, and the baseline plus 32 basins in the other cluster. Subplots (a), (b), (c), and (d), (e), (f) correspond to forecasts of 1, 2, and 3 days respectively.

surficial geology, or the presence of hydraulic structures such as dams. While the LSTM should be able to distinguish between these two basins using the one-hot encoded basin labels, incomplete explanatory variables may inhibit the ability of models to

350 transfer learned behaviour between basins.

Another explanation is that hydrologically similar basins contain a high degree of overlapping input-output patterns, reducing the amount of new information that can benefit predictions in the region of interest, in contrast to dissimilar basins. Information from dissimilar basins could be useful from a hydrological perspective, or simply provide a regularisation effect to the LSTM. Adding basins with distinct hydrological properties from some region of interest might occupy more neural pathways during

model training compared to basins that have similar properties, which could mitigate overfitting to the region of interest. However, this explanation is not supported by the fact that constraining the number of cells in the network did not produce comparable regularisation.

A final explanation is that while similar basins provide examples of similar hydrological behaviour, dissimilar basins provide examples of what not to predict. A simple analogy from image classification, is that a model trained to classify photos of dogs

might benefit more from being trained to some photos of cats, than to be trained to more photos of dogs.

Lastly, to better understand the effect that additions of similar and dissimilar basins have on model performance, we consider small, incremental additions. First, we begin with a model trained to 4 basins in a given cluster. Next, we consider two additions to the training dataset: 4 C0 basins, and 4 C1 basins. The addition that produces the best performance is retained, and the process is repeated. The models are retrained from scratch for each modification to the training dataset. The outcome is shown in Fig.

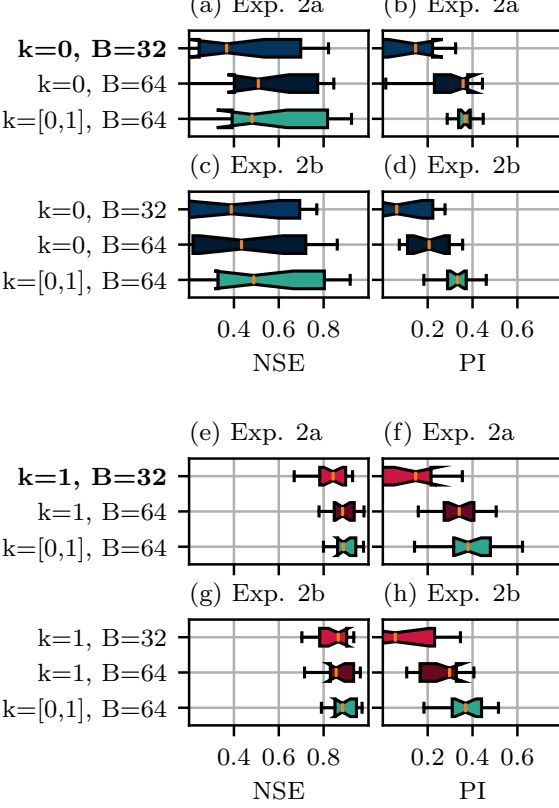

**Figure 10.** Boxplots showing the NSE and PI of models evaluated on 32 C0 basins for experiment 2a (a, b), and 2b (b, c), and 32 C1 basins for experiment 2a (e, f) and 2b (g, h). for a lead time of 3 days. The baseline model, which is trained uniquely using the evaluation basins, is indicated in bold. Blue, red, and green colours indicate models trained on C0, C1, and both types of basins.

11 for models evaluated on 4 C0 basins (a) and 4 C1 basins (b). In both cases, the models benefit from the addition of 4 basins that belong to the same cluster, however, afterwards there are no clear preference in terms of which basin clusters produce the best improvements. Despite some incremental additions hampering model performance, the performance on each evaluation improves across the larger training dataset, but the improvements decay exponentially, which is consistent with other studies that have looked at model performance across increasing training data (Kratzert et al., 2024; Gauch et al., 2021b). In Fig.

11(a), adding 4 similar basins to the 8 basin training dataset produces a large loss in performance, which highlights the lack of robustness against of models trained to very small basin sets. Between C0 and C1 evaluations, C0 basins are more sensitive to new training data, and are comparatively more likely than C1 basins to exhibit worse performance after the addition of new training basins. Since 4 basins is a relatively small sample size, the experiment was repeated with 4 different basins in each cluster, producing similar results, which are shown in Fig. S4.

The experimental results detailed above all assert the importance of hydrologically diverse, information rich training datasets. This is of particular importance in small regions of interest, where far away, dissimilar basins may be seen as not relevant to

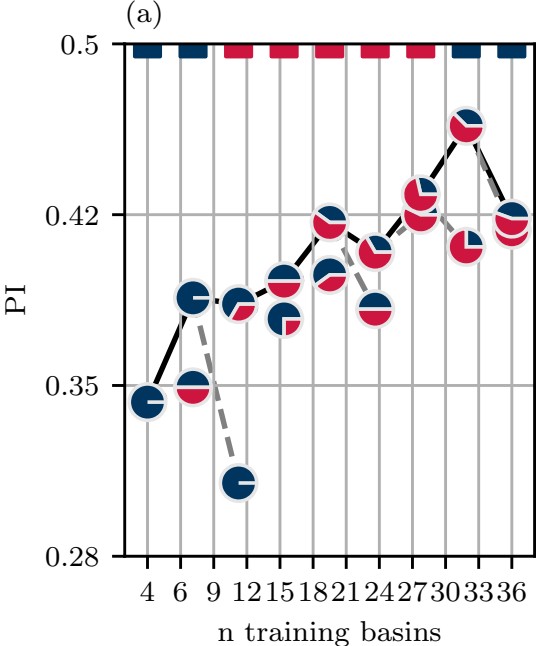
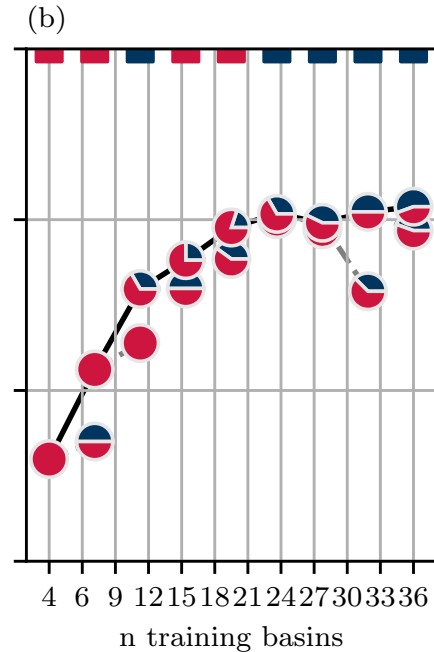

**Figure 11.** Model performance (PI) across increasing numbers of training basins for models evaluated on 4 C0 basins (a) and 4 C1 basins (b). Pie chart markers illustrate the proportion of C0 and C1 basins used in each training dataset. The coloured dashes along the top of each subplot indicate the which cluster produced the better addition to the training dataset.

the training task. This study presents many opportunities for future work on curating datasets for hydrological models. While our study uses a simple clustering approach to quantify hydrological diversity, more sophisticated approaches, such as one based on mutual information, may further improve the results. Additionally, relatively little work exists on transferring models

between different hydrological regions - which can potentially provide an improved starting point for model training, leading to better performance and more reliable FEWS.

## 4   Conclusions

The selection of training data is amongst the most important factors contributing to the performance of streamflow forecasting models. Our study showed that the performance of flow forecasting models relies on diverse training data, using a novel

use of cluster-based resampling to identify and maximise temporal and spatial hydrological diversity within training datasets. In the first set of experiments, cluster-based undersampling was used to eliminate redundant temporal data from training datasets, drastically reducing the computational demand of model training. The next set of experiments showed how, given some region of interest, data from hydrologically dissimilar basins can be much more useful than data from similar basins. This result is counter to the intuitive approach to curating training basins for training, which is to train models to a group of

hydrologically similar, or proximal basins. This outcome also highlights the need for large and hydrologically diverse training datasets. The latter can be combined with cluster-based temporal undersampling to generate diverse training sets that produce more performative models, for a fixed number of training observations, compared to models trained without cluster-based undersampling. Finally, temporal and spatial undersampling routines are combined to demonstrate how, for a fixed number of training samples, spatial hydrological diversity is much more beneficial than temporal diversity. These findings are critical to

improving the reliability and accuracy of flood forecasting models, and minimising the effects of flooding.

*Data availability.* The data used in this research is described in the publication by Arsenault et al. (2020) and availble for download through the Center for Open Science. Basemaps for several figures were provided by ESRI through the Contextily Python module.

## Funding

This work was supported by funding from York University, the Lassonde School of Engineering, and the Natural Sciences and

400 Engineering Research Council of Canada (grant no. RGPIN-2023-05077 and PGS-D scholarship).

*Competing interests.* The authors declare that they have no known competing financial interests or personal relationships that could have appeared to influence the work reported in this paper.

*Disclaimer.* The views expressed in this paper are those of the authors and do not necessarily reflect the views or policies of the affiliated organisations.

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

*Author contributions.* **E. Snieder:** conceptualisation; code; formal analysis; visualisation; writing - original draft. **U. T. Khan:** conceptualisation; funding acquisition; supervision; writing - editing, revisions.

*Competing interests.* The authors declare that they have no known competing financial interests or personal relationships that could have appeared to influence the work reported in this paper.

*Disclaimer.* The views expressed in this paper are those of the authors and do not necessarily reflect the views or policies of the affiliated organisations.

*Acknowledgements.* The authors would like to thank the HC3 research group at École de technologie supérieure for assembling the HYSETS database, which facilitated this work and other large-sample studies.

**Supplementary information**

**Static attribute statistics**

Attributes from Arsenault et al. (2020) are included directly in the dataset, whereas attributes from Addor et al. (2017) and MacKinnon (2010) are calculated from the hydrometeorological timeseries data in HYSETS. Seasonality attributes are calculated for flow and temperature using an additive seasonal decomposition for a frequency of one year. The seasonality is then given as:

$$\text{Seasonality} = 1 - \frac{\sum(w_t - \overline{w_t})^2}{\sum((w_t + s_t) - \overline{(w_t + s_t)})^2} \tag{1}$$

where $s_t$ and $w_t$ are the seasonal and residual (i.e. white noise) terms of additive seasonal decomposition. Negative seasonality values are truncated to 0, resulting in seasonality values ranging between 0 and 1, corresponding to no and high seasonality.

**Table S1.** Basin static attribute statistics (mean, standard deviation, 10th, 50th, and 90th percentile) calculated for 2363 Canadian catchments.

| | mean | std | 10% | 50% | 90% | source |
|---|---|---|---|---|---|---|
| elevation (m) | 703 | 740 | 93.5 | 380 | 1.89e+03 | (Arsenault et al., 2020) |
| slope (deg) | 6.12 | 6.52 | 0.596 | 3.02 | 16 | (Arsenault et al., 2020) |
| gravelius (-) | 1.69 | 0.359 | 1.32 | 1.61 | 2.17 | (Arsenault et al., 2020) |
| aspect (deg) | 172 | 89 | 59.2 | 162 | 302 | (Arsenault et al., 2020) |
| land use forest (frac) | 0.435 | 0.294 | 0.0269 | 0.45 | 0.826 | (Arsenault et al., 2020) |
| land use grass (frac) | 0.0843 | 0.143 | 0.0018 | 0.027 | 0.24 | (Arsenault et al., 2020) |
| land use wetland (frac) | 0.0448 | 0.0892 | 0 | 0.008 | 0.145 | (Arsenault et al., 2020) |
| land use water (frac) | 0.0142 | 0.0315 | 0 | 0.0036 | 0.0349 | (Arsenault et al., 2020) |
| land use urban (frac) | 0.0945 | 0.167 | 0.0011 | 0.0464 | 0.219 | (Arsenault et al., 2020) |
| land use shrubs (frac) | 0.0987 | 0.153 | 0.0001 | 0.0336 | 0.293 | (Arsenault et al., 2020) |
| land use crops (frac) | 0.223 | 0.284 | 0 | 0.0754 | 0.745 | (Arsenault et al., 2020) |
| land use snow ice (frac) | 0.00485 | 0.0325 | 0 | 0 | 0.0001 | (Arsenault et al., 2020) |
| permeability (log; $m^2$) | -13.9 | 1.18 | -15.4 | -14 | -12.3 | (Arsenault et al., 2020) |
| porosity (frac) | 0.123 | 0.0641 | 0.0286 | 0.124 | 0.205 | (Arsenault et al., 2020) |
| precip mean (mm/d) | 2.74 | 1.19 | 1.29 | 2.71 | 3.93 | (Addor et al., 2017) |
| precip high (mm/d) | 13.7 | 5.94 | 6.44 | 13.6 | 19.6 | (Addor et al., 2017) |
| precip low (mm/d) | 1 | 0 | 1 | 1 | 1 | (Addor et al., 2017) |
| precip mean (mm/mo) | 83.2 | 36.2 | 38.6 | 82.5 | 120 | (Addor et al., 2017) |
| precip mean (mm/y) | 998 | 435 | 464 | 990 | 1.43e+03 | (Addor et al., 2017) |
| precip high freq (d/yr) | 3.93 | 0.528 | 3.21 | 4.06 | 4.46 | (Addor et al., 2017) |
| precip high dur (d) | 1.24 | 0.131 | 1.12 | 1.2 | 1.42 | (Addor et al., 2017) |
| precip low freq (d/yr) | 48.6 | 5.82 | 41.2 | 49 | 55.7 | (Addor et al., 2017) |
| precip low dur (d) | 5.5 | 2.78 | 3.53 | 4.65 | 8.26 | (Addor et al., 2017) |
| baseflow index (-) | 0.646 | 0.168 | 0.416 | 0.665 | 0.846 | (Addor et al., 2017) |
| q mean (mm/d) | 1.3 | 1.38 | 0.151 | 0.98 | 2.56 | (Addor et al., 2017) |
| q high (mm/d) | 11.7 | 12.5 | 1.36 | 8.82 | 23 | (Addor et al., 2017) |
| q low (mm/d) | 0.261 | 0.277 | 0.0302 | 0.196 | 0.512 | (Addor et al., 2017) |
| q high freq (d/yr) | 95.8 | 122 | 1 | 49 | 268 | (Addor et al., 2017) |
| q high dur (d) | 2.63 | 4.13 | 1 | 1.74 | 5.12 | (Addor et al., 2017) |
| q low dur (d) | 26.8 | 44.9 | 3.56 | 15.4 | 56.8 | (Addor et al., 2017) |
| q zero freq (d/yr) | 351 | 1.43e+03 | 0 | 0 | 698 | (Addor et al., 2017) |
| q 95 (mm/d) | 4.43 | 4.52 | 0.429 | 3.41 | 8.81 | (Addor et al., 2017) |
| q 5 (mm/d) | 0.162 | 0.316 | 0 | 0.0722 | 0.389 | (Addor et al., 2017) |
| runoff ratio (-) | 0.45 | 0.374 | 0.101 | 0.371 | 0.842 | (Addor et al., 2017) |
| q adf (-) | -12.4 | 6.99 | -19.4 | -11.6 | -5.16 | (MacKinnon, 2010) |
| q seasonality (-) | 0.272 | 0.231 | 0.0543 | 0.192 | 0.65 | Eqn. 1 |
| tmean seasonality (-) | 0.782 | 0.0633 | 0.724 | 0.795 | 0.831 | Eqn. 1 |
| tmean (c) | 9.95 | 5.32 | 3.77 | 9.81 | 17.5 | - |
| tmax annual mean (c) | 34.6 | 3.05 | 30.8 | 35 | 38.1 | - |
| tmin annual mean (c) | -20.8 | 9.73 | -33 | -21.3 | -7.76 | - |
| q var $(mm/d)^2$ | 9.5 | 27.3 | 0.129 | 3.5 | 19 | - |

**Table S2.** Basin static attribute statistics (mean, standard deviation, 10th, 50th, and 90th percentile) calculated for 2363 Canadian catchments.

| Dynamic attributes | Source | Units |
|---|---|---|
| Discharge | Water Survey of Canada | mm (basin normalised) |
| Precipitation | Environment and Climate Change Canada (Quality controlled) | mm |
| Temperature | Environment and Climate Change Canada (Quality controlled) | degC |
| SWE | ERA5 | mm |

## Runtimes for experiments 1a and 1b

Runs were performed on a workstation with an AMD Ryzen 5600G (CPU learning) and 32GB DDR4 RAM.

**Table S3.** Runtimes for experiment 1a: cluster-based temporal undersampling. The symbol '-' denotes a compromised runtime calculation.

| configuration | run duration (minutes) |
|---|---|
| baseline | 116.37 |
| $CUS_Q$ (K=12, $\phi$=0.50) | - |
| $CUS_Q$ (K=6, $\phi$=0.25) | 37.70 |
| $CUS_Q$ (K=6, $\phi$=0.50) | 73.33 |
| RUS ($\phi$=0.25) | 57.55 |
| RUS ($\phi$=0.50) | 62.68 |

**Table S4.** Runtimes for experiment 1b: cluster-based spatial undersampling. The symbol '-' denotes a compromised runtime calculation.

| configuration | run duration (minutes) |
|---|---|
| baseline | 176.65 |
| $CUS_B$ (K=2, B=64) | 94.60 |
| $CUS_B$ (K=32, B=64) | 76.33 |
| $CUS_B$ (K=32, B=96) | 108.95 |
| $CUS_B$ (K=8, B=64) | 90.75 |
| RUS (B=64) | 66.43 |
| RUS (B=96) | - |

**Sensitivity of silhouette scores to minimum cluster size**

The clustering outcome and silhouette scores are highly dependent on the minimum cluster size. Unconstrained k-means clustering results in an optimum K value of 8, however, some of these clusters do not contain enough basins to use for validation. A comparison of silhouette scores across the numbers of clusters for minimum cluster sizes of 2, 8, 16, 32, 64, is provided in Fig. 1 below. Although greater numbers of clusters produce more hydrologically diverse sets of basins, there are simply not enough basins in each cluster to form a training dataset with balanced hydrological conditions (i.e., an equal number of basins from each cluster).

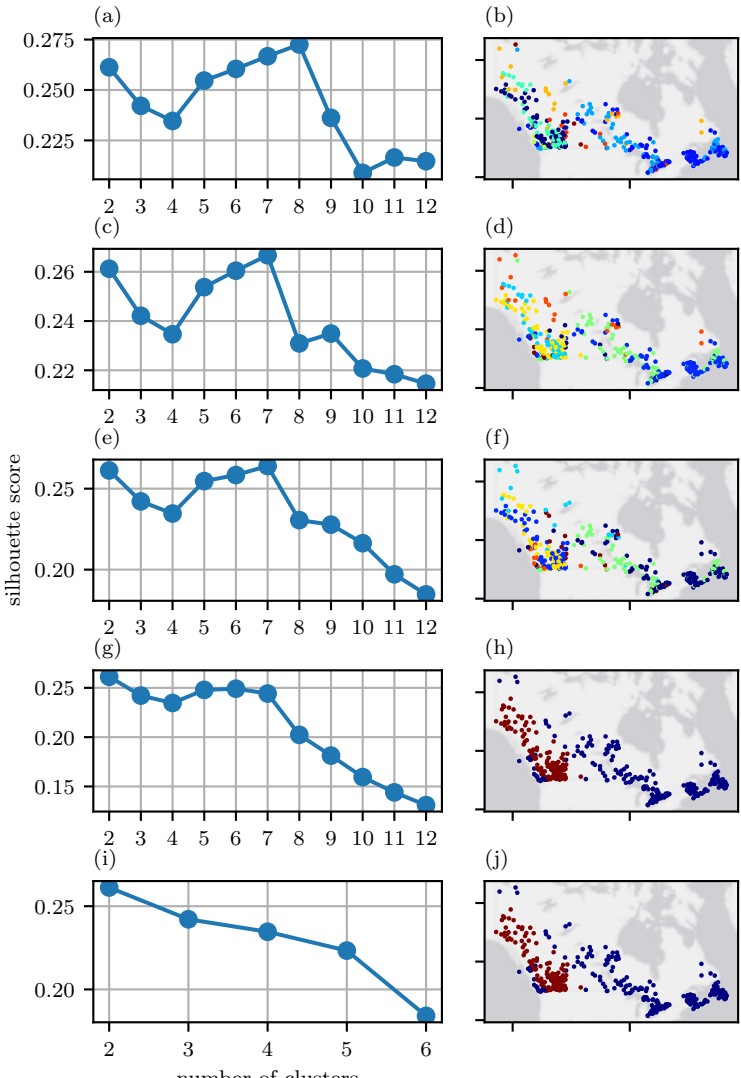

**Figure S1.** Silhouette scores (left) for minimum cluster sizes of 2 (a), 8 (c), 16 (e), 32 (g), and 64 (i). Respective clustering results (right) for K values (corresponding to maximum silhouette score) of 8 (b), 7 (d, f), and 2 (h, j). World Gray Canvas basemap provided by ESRI.

## Two-cluster attributes

**Table S5.** Mean and standard deviation of basin attributes for 2 cluster configuration.

|  | C0 mean | C0 stdev | C1 mean | C1 stdev |
|---|---|---|---|---|
| elevation (m) | 307.05 | 179.09 | 1432.84 | 327.84 |
| slope (deg) | 3.11 | 3.82 | 16.15 | 6.84 |
| gravelius (-) | 2.11 | 0.5 | 1.86 | 0.37 |
| aspect (deg) | 155.91 | 88.78 | 169.09 | 100.34 |
| land use forest (frac) | 0.58 | 0.27 | 0.6 | 0.2 |
| land use grass (frac) | 0.04 | 0.08 | 0.2 | 0.13 |
| land use wetland (frac) | 0.04 | 0.07 | 0.0 | 0.0 |
| land use water (frac) | 0.06 | 0.07 | 0.02 | 0.02 |
| land use urban (frac) | 0.04 | 0.1 | 0.01 | 0.01 |
| land use shrubs (frac) | 0.09 | 0.11 | 0.09 | 0.05 |
| land use crops (frac) | 0.14 | 0.26 | 0.02 | 0.11 |
| land use snow ice (frac) | 0.0 | 0.01 | 0.05 | 0.1 |
| permeability (log; $m^2$) | -14.19 | 1.17 | -14.07 | 0.9 |
| porosity (frac) | 0.1 | 0.06 | 0.11 | 0.06 |
| precip mean (mm/d) | 2.68 | 1.23 | 2.14 | 1.43 |
| precip high (mm/d) | 13.39 | 6.15 | 10.69 | 7.14 |
| precip low (mm/d) | 1.0 | 0.0 | 1.0 | 0.0 |
| precip mean (mm/mo) | 81.2 | 37.74 | 64.45 | 43.81 |
| precip mean (mm/y) | 974.38 | 452.9 | 773.45 | 525.76 |
| precip high freq (d/yr) | 3.57 | 0.5 | 3.33 | 0.66 |
| precip high dur (d) | 1.16 | 0.11 | 1.23 | 0.07 |
| precip low freq (d/yr) | 45.0 | 5.12 | 45.65 | 6.79 |
| precip low dur (d) | 3.99 | 1.63 | 4.64 | 0.98 |
| baseflow index (-) | 0.71 | 0.13 | 0.78 | 0.1 |
| q mean (mm/d) | 1.8 | 1.54 | 1.86 | 1.79 |
| q high (mm/d) | 16.18 | 13.88 | 16.77 | 16.13 |
| q low (mm/d) | 0.36 | 0.31 | 0.37 | 0.36 |
| q high freq (d/yr) | 69.38 | 91.85 | 47.28 | 83.23 |
| q high dur (d) | 2.67 | 2.69 | 2.35 | 2.54 |
| q low dur (d) | 29.66 | 34.52 | 42.33 | 35.39 |
| q zero freq (d/yr) | 183.47 | 970.46 | 135.96 | 1064.88 |
| q 95 (mm/d) | 6.13 | 5.27 | 6.42 | 5.57 |
| q 5 (mm/d) | 0.21 | 0.27 | 0.24 | 0.29 |
| runoff ratio (-) | 0.6 | 0.38 | 0.87 | 0.57 |
| q adf (-) | -14.24 | 5.33 | -11.23 | 4.58 |
| q seasonality (-) | 0.3 | 0.2 | 0.51 | 0.27 |
| tmean seasonality (-) | 0.8 | 0.1 | 0.76 | 0.11 |
| tmean (c) | 3.69 | 3.81 | 3.31 | 3.94 |
| tmax annual mean (c) | 31.03 | 2.27 | 31.17 | 2.83 |
| tmin annual mean (c) | -30.24 | 9.02 | -31.26 | 10.03 |
| q var $(mm/d)^2$ | 11.02 | 26.07 | 10.2 | 21.08 |

**Additional two-cluster comparison performance metrics**

The NSE $\alpha$ and $\beta$ decompositions are taken from (Gupta et al., 2009):

$$\alpha - \text{NSE} = \sigma_{\hat{q}} - \sigma_q \tag{2}$$

where $\sigma_{\hat{q}}$ and $\sigma_q$ are the standard deviations of the simulated and observed streamflow.

$$\beta - \text{NSE} = \frac{\mu_{\hat{q}} - \mu_q}{\sigma_q} \tag{3}$$

where $\mu_{\hat{q}}$ and $\mu_q$ are the means of the simulated and observed streamflows.

The overfitting ratio (OFR), which quantifies the relative difference in train and test error, is calculated based on the MSE:

$$\text{MSE} = \frac{1}{n} \sum (q_t - \hat{q}_t)^2 \tag{4}$$

The OFR is given as:

$$\text{OVERFITTING RATIO} = 1 - \frac{\text{MSE}_{\text{test}}}{\text{MSE}_{\text{train}}} \tag{5}$$

where $\text{MSE}_{\text{test}}$ and $\text{MSE}_{\text{train}}$ are the MSE scores on the test, and train partitions, respectively. The OFR ranges $-\infty$ to an ideal value of 1, which corresponds to 0 error on the test dataset. Scores of 0 indicate the same error on train and test data.

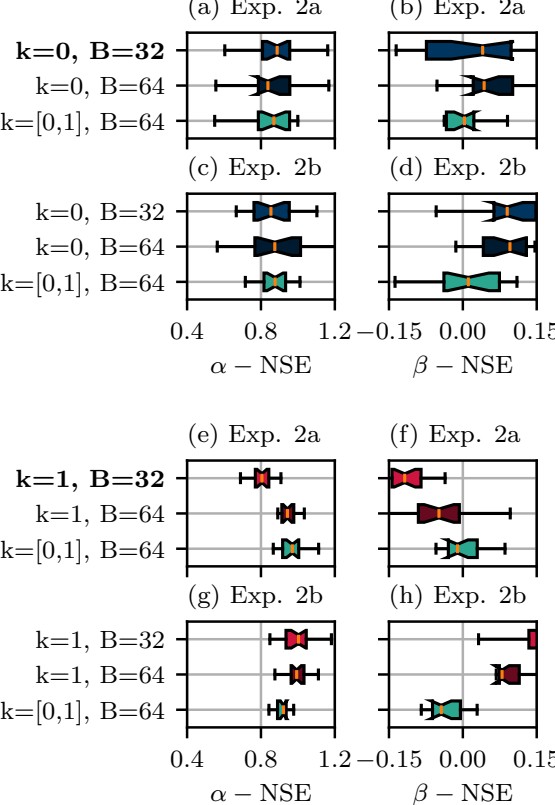

**Figure S2.** Boxplots showing the $\alpha - $NSE (left) and $\beta - $NSE (right) of models evaluated on 32 C0 basins for experiment 2a (a, b), and 2b (b, c), and 32 C1 basins for experiment 2a (e, f) and 2b (g, h). for a lead time of 3 days. The baseline model, which is trained uniquely using the evaluation basins, is indicated in bold. Blue, red, and green colours indicate models trained on C0, C1, and both types of basins.

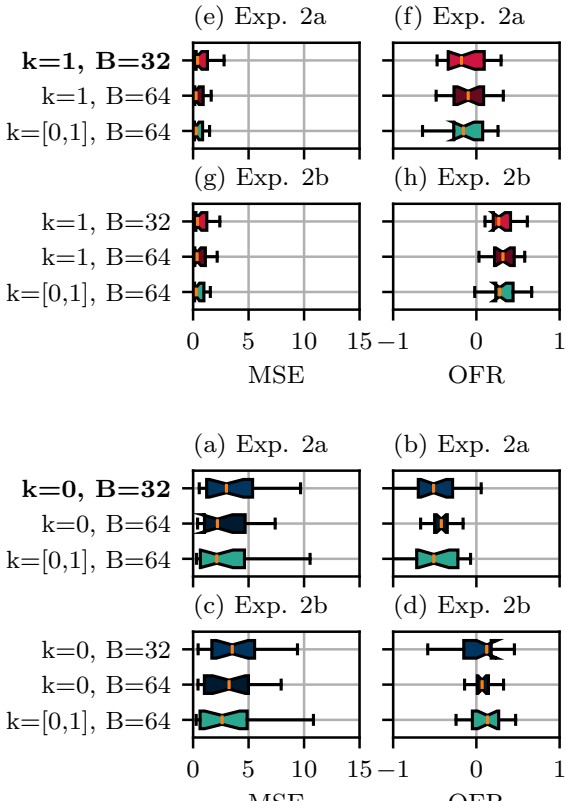

**Figure S3.** Boxplots showing the MSE (left) and OFR (right) of models evaluated on 32 C0 basins for experiment 2a (a, b), and 2b (b, c), and 32 C1 basins for experiment 2a (e, f) and 2b (g, h). for a lead time of 3 days. The baseline model, which is trained uniquely using the evaluation basins, is indicated in bold. Blue, red, and green colours indicate models trained on C0, C1, and both types of basins.

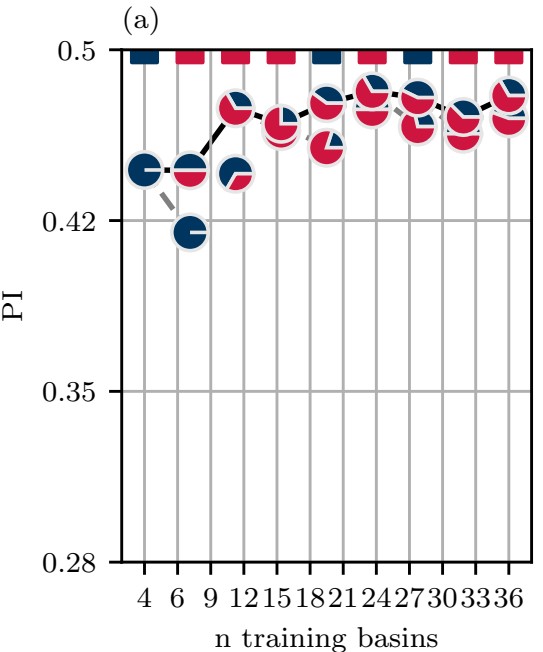 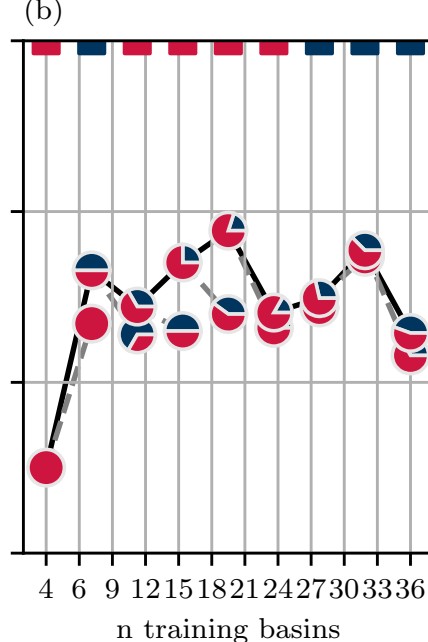

**Figure S4.** Model performance (PI) across increasing numbers of training basins for models evaluated on 4 C0 basins (a) and 4 C1 basins (b). Pie chart markers illustrate the proportion of C0 and C1 basins used in each training dataset. The coloured dashes along the top of each subplot indicate the which cluster produced the better addition to the training dataset.