# Peer review of "A diversity centric strategy for the selection of spatio-temporal training data for LSTM-based streamflow forecasting"

_Hydrology and Earth System Sciences, 2024_

## Author Comment (AC1)

**Supplementary information for RC1 response.**

[Figure]

*Figure RC1a: Revised Figure 3 with reduced marker size.*

[Figure]

**Figure RC1b: revised Figure 4 with reduced linewidths.**

[Figure]

**Figure RC1c: revised Figure 5 with reduced linewidths.**

---

## Author Comment (AC2)

**Supplementary information for RC2 response.**

**Runtimes for experiments 1a and 1b**

Runs were performed on a workstation with an AMD Ryzen 5600G (CPU learning) and 32GB DDR4 RAM.

**Table S2.** Runtimes for experiment 1a: cluster-based temporal undersampling. The symbol '-' denotes a compromised runtime calculation.

| configuration | run duration (minutes) |
|---|---|
| baseline | 116.37 |
| $CUS_Q$ (K=12, $\phi$=0.50) | - |
| $CUS_Q$ (K=6, $\phi$=0.25) | 37.70 |
| $CUS_Q$ (K=6, $\phi$=0.50) | 73.33 |
| RUS ($\phi$=0.25) | 57.55 |
| RUS ($\phi$=0.50) | 62.68 |

*Table RC2a: runtimes for experiment 2a.*

**Table S3.** Runtimes for experiment 1b: cluster-based spatial undersampling. The symbol '-' denotes a compromised runtime calculation.

| configuration | run duration (minutes) |
|---|---|
| baseline | 176.65 |
| $CUS_B$ (K=2, B=64) | 94.60 |
| $CUS_B$ (K=32, B=64) | 76.33 |
| $CUS_B$ (K=32, B=96) | 108.95 |
| $CUS_B$ (K=8, B=64) | 90.75 |
| RUS (B=64) | 66.43 |
| RUS (B=96) | - |

*Table RC2b: runtimes for experiment 2b.*

[Figure]

**FigureRC2a: Hyperparameter optimisation results in terms of PI performance on the validation dataset.**

[Figure]

Figure RC2b: revised Figure 4 with reduced linewidths.

[Figure]

**Figure S1.** Silhouette scores (left) for minimum cluster sizes of 2 (a), 8 (c), 16 (e), 32 (g), and 64 (i). Respective clustering results (right) for K values (corresponding to maximum silhouette score) of 8 (b), 7 (d, f), and 2 (h, j). World Gray Canvas basemap provided by ESRI.

***Figure RC2c: Relationship between silhouette score and minimum cluster size.***

[Figure]

**Figure RC2d: Updated Figure 10 with an increased number of subplots and detailed subplot titles.**

[Figure]

[Figure]

*Figure RC2e: Original experimental results.*

[Figure]

[Figure]

*FigureRC2f: Additional experimental run on two sets of 4 different basins*

---

## Author Response (AR1)

**Reviewer 1 feedback**

Review of "A diversity centric strategy for the selection of spatio-temporal training data for LSTMI-based streamflow forecasting" by Everett Snieder and Usman T. Khan

This study investigates the impact of forming hydrologically diverse training datasets on model performance and generalization. It aims to quantify hydrological diversity using clustering and to evaluate the effects of adding similar and dissimilar basins to training datasets. The study successfully achieves its objective by demonstrating the importance of diverse training data in improving model performance and generalization.

The study's findings and discussions offer meaningful implications for future research and model training practices, making it a useful contribution to hydrology.

The paper is well written, structured and clear.

Thank you for your encouraging comments and thoughtful feedback. We greatly appreciate the time and effort you took to share your perspectives with us. We hope that the detailed response provided below meets your expectations and addresses all of your concerns.

**General comments:**

R1-1: The rationale for choosing the specific dynamic features is not entirely clear to me. I would appreciate if the authors could slightly more elaborate on this.

Assuming the dynamic features are those used for temporal clustering; these five include streamflow, day of year of sample (2 features), and change in flow (2 features). They were based on expert knowledge and features used in previous studies:

Day of year attributes can be used to allow the clustering model to distinguish between different seasonal periods (Abrahart et al., 2001) and sequential flows, which provides the model with change in flow (Toth, 2009).

These details are included in the revised version of the manuscript, where we introduce the dynamic features for Exp 1a (L185-189):

> "The engineered feature set includes streamflow ($q_t$), two streamflow gradient features (given as ($q_{t-3} - q_t$)/3 and $q_{t-1} - q_t$), and two day of year features (given as sin ($2\pi t/365$) and cos ($2\pi t/365$) where t is the day of year 1,...,365. While the day of year is discontinuous between 365 and 1, the sin and cos decomposition offers a continuously changing pairing across each year, which increases the likelihood of clusters spanning from December to January. The streamflow gradient encourages the representation of rising and falling limbs within the clusters, which are not distinguishable using only the streamflow state"

Abrahart, R. J., See, L., and Kneale, P. E.: Investigating the role of saliency analysis with a neural network rainfall-runoff model, Computers & Geosciences, 27, 921–928, https://doi.org/10.1016/S0098-3004(00)00131-X, 2001.

Toth, E.: Classification of hydro-meteorological conditions and multiple artificial neural networks for streamflow forecasting, Hydrology and Earth System Sciences, 13, 1555–1566, https://doi.org/10.5194/hess-13-1555-2009, 2009.

R1-2: In the methodology, the clustering of catchment static attributes and hydrological dynamics is done independently. Can the authors clarify the motivation for this? What would happen if this were done in one step?

The motivation for clustering basins and dynamic features independently was (1) to evaluate them independently from one another, (2) it ensured that an even number of samples would be drawn from each basin, which is simplifies some of our experiments, which compare varying quantities of training data from hydrologically dissimilar basins. A one-step unified approach would be very interesting and potentially useful in selecting training data, but we did not deem it necessary to meet our research objectives. We believe that a one-step clustering method would have a similar effect to clustering in series. The motivation for the two-step approach is included in the revised manuscript (218-220):

> "Note that the two clustering applications described above are standalone and can be combined by using them in series. While a unified, spatiotemporal clustering method could be used to cluster samples in a single step, separating spatial and temporal clustering allows for each method to be assessed independently of one another."

- The authors mention that regulated catchments may behave differently than expected from the selected catchment attributes. How many of the catchments are regulated? Is there a catchment attribute that accounts for this and would it be worth adding such? In the authors' opinion, would it affect the equal sample size, which was one of the constraints for the clustering approach?

Unfortunately, there are no catchment attributes within HYSETS that indicate whether a catchment is regulated and the number of regulated catchments is not known. While there is an increasing number of hydraulic infrastructure databases (Zhang et al., 2023; Mulligan et al., 2020), they are incomplete and lack key information such as year of construction. The effects of infrastructure on regional learning and spatial generalisation is a very interesting topic and worthy of its own study.

Filtering for regulated catchments would certainly reduce the number of available basins and potentially bias the geographical distributions of basins (e.g., Quebec and British Columbia have high numbers of regulated catchments).

We can include a short discussion of this topic in the revised manuscript (L110-112):

"Unfortunately, there are no attributes that reveal whether a catchment is regulated by built infrastructure. However, the model may still be able to learn the effects of built infrastructure implicitly, through the rainfall-runoff relationship."

Mulligan, M., van Soesbergen, A., and Sáenz, L.: GOODD, a global dataset of more than 38,000 georeferenced dams, Sci Data, 7, 31, https://doi.org/10.1038/s41597-020-0362-5, 2020.

Zhang, A. T. and Gu, V. X.: Global Dam Tracker: A database of more than 35,000 dams with location, catchment, and attribute information, Sci Data, 10, 111, https://doi.org/10.1038/s41597-023-02008-2, 2023.

R1-3: Although the focus of the study is on LSTM-based streamflow forecasting, there is a body of literature on the value of data for calibration, regionalization and on model testing that I would consider relevant in this context. This starts, for example, with Klemes (KLEMEŠ, V. (1986). Operational testing of hydrological simulation models. Hydrological Sciences Journal, 31(1), 13–24. https://doi.org/10.1080/02626668609491024)

Thank you for the recommended article. Our submitted manuscript uses the split-sample approach described in Klemes (1986) and have added the citation in-text (L117-119):

"Basin are removed if they have less than 80% data availability within any of the training, validation, and testing periods, which span a total of 36 years from Oct. to Sep. of 1982-1994, 1994-2006 and 2006-2018, respectively, following the split-sample method (KLEMEŠ, 1986)."

R1-4: Some of the figures have linewidth that are too big for the plot size and make them difficult to read (Figure3, Figure4 (mainly upper panel), Figure5) Consider making the ratio more readable.

We apologise for the lack of clarity in the figure and have reduced the linewidth in the plots. The updated figures with the thinner linewidths are copied below, and are included in the revised manuscript. See the revised Figures RC1a, RC1b, and RC1c below, and in the updated manuscript (L273, L291, and L306).

[Figure]

*Figure RC1a: Revised Figure 3 with reduced marker size.*

[Figure]

*Figure RC1b: revised Figure 4 with reduced linewidths and lead times in titles.*

[Figure]

*FigureRC1c: revised Figure 5 with reduced linewidths.*

**Detailed comments:**

R1-5: L40 I would already here add the references for the mentioned CAMELS data sets

Thank you for this recommendation. We've replaced the mentions of CAMELS with a single reference and citation for Caravan, which includes all the CAMELS datasets (L34-41):

"Often times, models are trained to complete large-sample datasets such as those included in Caravan (Kratzert et al., 2023). However, with increasingly large, global hydrological datasets, it is not always practical or feasible to train to all available data, especially when conducting computationally expensive tasks such as hyperparameter selection, which is required to achieve optimum model performance, or creating multimodel ensembles. Therefore, there is a need for improved guidance on efficient methods for training data selection, to maximise model performance and generalisation. Many of the deep learning advances in hydrology (e.g., (Kratzert et al., 2019a), (Klotz et al., 2022), Lees et al. (2022), and (Gauch et al., 2021a)) have utilised well established large-sample basins (see Caravan (Kratzert et al., 2023) and the datasets therein)."

R1-6: L48 References to these many studies? or say that they are mentioned below

We apologise for the lack of clarity and have modified the text as follows (L48):

"Many studies, which are reviewed below, have applied clustering to spatial and temporal data as a means to quantify hydrological diversity."

R1-7: L52 SOM is the first time mentioned, please introduced the full term

SOM refers to a self-organising map; we apologise for the omission and have updated the manuscript to reflect this change (L53-54):

> "Anctil and Lauzon (2004) apply a self-organising map (SOM) to streamflow data in a single basin to create a training dataset with a balanced representation of diverse hydrological states"

R1-8: L56 Direct citation

We apologise for this error and have fixed the reference (L57):

> "Snieder et al. (2021) partition streamflows into typical streamflows and high streamflows"

R1-9: L66 all available basins

Thank you for identifying this error – we have made the correction (L67):

> "finding that a model trained to all available basins typically …"

R1-10: L62-79 Here it would be good to see if different clustering approaches were used in the mentioned studies or if all used k-means and for which motivation

Thank you for this recommendation, we have added detail about the classification methods used in each study. The updated text is as follows (L64-75):

> "However, other studies have used clustering to estimate hydrological diversity, such that basin selection can explicitly account for hydrological diversity. These cases tend to use some form of clustering (either supervised or unsupervised) to quantify hydrological diversity within training data and the effects it has on model generalisation. Zhang et al. (2022) applied K-means clustering to a set of 35 mountainous basins in China based on hydroclimatic attributes, finding that a model trained to all available basins typically outperformed those trained to individual clusters. Hashemi et al. (2022) applied a similar approach by classifying basins into distinct hydrological regimes based hydrometeorological thresholds. As done in (Zhang et al., 2022), their study compared locally and globally trained models, finding only minor differences in the performance between the two. A common problem in comparing global and locally trained models is that these comparisons typically do not control for sample size. As a result, the improved performance of the global model can be impacted by the regularisation effect on the sample size. In other words, deep learning models trained to small datasets may be overfitted and thus, poorly generalised. Fang et al. (2022) accounts for this potential issue. Their study used existing `ecoregion' basin classification, which were classified by the United States Environmental Protection Agency, and evaluates the effects of additional training basins at three similarity intervals."

R1-11: L109 basin label

Thank you for identifying this error – we have made the correction (L112):

"Lastly, the input feature set also contains one-hot encoded basin label (Lees et al., 2022)"

R1-12: L131 , -> .
Thank you for identifying this error – we have made the correction (L134-136):

"While an input sequence of 365 days is commonly used for streamflow prediction (Kratzert et al., 2019b; Arsenault et al., 2023b). Gauch et al. (2021b) noted that small sequences are better suited to small basin sets, and have been used in AR models (Nevo et al., 2022)."

R1-13: Figure 2 caption add .
Thank you for identifying this error – we have made the correction (L266-267):

"Figure 2. Scatterplot matrix illustrating the temporal clustering results for Basin 01AD003 streamflows (K=6). Markers are colourised by cluster and kernel smoothed histograms are shown along the diagonal. Axis labels q, qgrd1, qgrd3, sin, and cos, are shortform for qt, (qt-3 - qt)/3, qt-1 - qt, sin-1 (t/365), and cos-1 (t/365), respectively."

R1-14: Figure 4 It would be helpful for reading the figure, if the last sentence of the caption could be included in the plot itself, for instance by using facets.

Thank you for this recommendation – we've added the lead time to the subplot titles, which is shown in Figure RC1b.

**Reviewer 2 feedback**

I agree with the authors that large-scale training of LSTM models can lead to performance trade-offs between different regions. However, a diverse training dataset is crucial for improving overall model performance. This study demonstrates that temporal undersampling does not compromise model accuracy and enhances model efficiency, while spatial undersampling results in only a marginal performance decrease. Surprisingly, the authors found that adding dissimilar basins leads to greater improvements than adding more similar basins. The paper provides a discussion on the underlying reasons for this observation. Overall, the paper is well-written and thoroughly demonstrated, and I recommend its publication with moderate revisions.

Thank you for your kind comments and valuable feedback. We hope that the comprehensive response below meets your expectations and thoroughly addresses any concerns you may have.

**General Comments:**

R2-1: I suggest mentioning the computational time used for the CUS, RUS, and baseline model training to highlight the practical benefits this undersampling method can bring to modeling.

Thank you for this recommendation, we have added training times to the SI. As shown in the table below, there are practical benefits to the proposed undersampling methods, in terms of reduced computational time. The runtimes are included in Tables RC2a and RC2b in the below, as well as in the revised manuscript (L283-286):

> "Reducing the computational requirement of HPO speeds up model development, or allows for more extensive HPO, potentially improving model accuracy, thus FEWS reliability. Improvements in runtimes, which are reported in Tables S3 and S4, were typically found to be proportional to undersampling rates".

**Runtimes for experiments 1a and 1b**

Runs were performed on a workstation with an AMD Ryzen 5600G (CPU learning) and 32GB DDR4 RAM.

**Table S3.** Runtimes for experiment 1a: cluster-based temporal undersampling. The symbol '-' denotes a compromised runtime calculation.

| configuration | run duration (minutes) |
|---|---|
| baseline | 116.37 |
| $CUS_Q$ (K=12, $\phi$=0.50) | - |
| $CUS_Q$ (K=6, $\phi$=0.25) | 37.70 |
| $CUS_Q$ (K=6, $\phi$=0.50) | 73.33 |
| RUS ($\phi$=0.25) | 57.55 |
| RUS ($\phi$=0.50) | 62.68 |

*Table RC2a: runtimes for experiment 2a.*

**Table S4.** Runtimes for experiment 1b: cluster-based spatial undersampling. The symbol '-' denotes a compromised runtime calculation.

| configuration | run duration (minutes) |
|---|---|
| baseline | 176.65 |
| $CUS_B$ (K=2, B=64) | 94.60 |
| $CUS_B$ (K=32, B=64) | 76.33 |
| $CUS_B$ (K=32, B=96) | 108.95 |
| $CUS_B$ (K=8, B=64) | 90.75 |
| RUS (B=64) | 66.43 |
| RUS (B=96) | - |

Table RC2b: runtimes for experiment 2b.

**Detailed Comments:**

R2-2: Line 52: What does SOM represent?

SOM refers to a self-organising map; we apologise for the omission and have updated the manuscript to reflect this change (L53-54):

> "Anctil and Lauzon (2004) apply a self-organising map (SOM) to streamflow data in a single basin to create a training dataset with a balanced representation of diverse hydrological states"

R2-3: Line 56: Please correct the citation format.

We apologise for this error and have fixed the reference (L57):

> "Snieder et al. (2021) partition streamflows into typical streamflows and high streamflows"

R2-4: Line 103-104: Which forcing dataset is used here?

The forcing dataset is obtained from the HYSETS (the Canadian portion of which is also included in Caravan). The text has been clarified as follows (L104-106):

> " Additional dynamic features from the HYSETS database includes daily basin-averaged minimum temperature, maximum temperature, precipitation, and snow water equivalent (SWE), which are listed in Table 2."

R2-5: Line 127-132: How are these hyperparameters tuned? For a dataset with 2000 basins, the batch size seems small.

Hyperparameters were tuned on the training dataset and validated on an independent partition of 12 years (L115), which is independent of the test partition reported in the manuscript. An ad-hoc grid-search (one hyperparameter modification at a time) was conducted using a model trained to 64 randomly sampled basins (rather than using the full dataset of 2500). Varying the batch size was not found to significantly impact model performance, as shown in Figure RC2a below.

[Figure]

FigureRC2a: Hyperparameter optimisation results in terms of PI performance on the validation dataset.

R2-6: Line 183: Please clarify what the two day of year features are. It is unclear.

We have rewritten and corrected a small error in the day of year feature description. The day of year features are calculated using the day of year (from 1 (Jan. 1) to 365 (Dec. 31)) and are used to allow the model to distinguish flows occurring in different seasons. Since the day of year value is not continuous from December to January, the day of year is decomposed into sin and cos components, which as a pair, provide continuous values across the year. Mathematically, these features are calculated as $sin((2/\pi)*(DOY/365))$ and $cos((2/\pi)*(DOY/365))$ where DOY is the day of year.

This section has been rewritten (L184-189):

> "The engineered feature set includes streamflow (qt), two streamflow gradient features (given as (qt-3 – qt)/3 and qt-1 – qt), and two day of year features (given as sin (2πt/365) and cos (2πt/365) where t is the day of year {1,...,365}. While the day of year is discontinuous between 365 and 1, the sin and cos decomposition offers a continuously changing pairing across each year, which increases the likelihood of clusters spanning from December to January. The streamflow gradient encourages the representation of rising and falling limbs within the clusters, which are not distinguishable using only the streamflow state."

R2-7: Line 234: Which method is used for clustering in experiment 2? Is it K-means?

Yes, K-means was used (Constrained K-means variant). We have added a reference to the corresponding description in the methods section (L241-242):

> "In experiment 2a, basins are divided into two clusters (which are referred to as C0 and C1), using the K-means method described in Sec. 2.5."

R2-8: Line 240: The configurations of experiment 2b are unclear to me. Please clarify.

We apologise for the lack of clarity. The objective of this experiment is to determine whether temporal and spatial undersampling can be used in series to further reduce the computational requirements of training. The configuration of experiment 2b mirrors that of experiment 2a, but cluster-based temporal undersampling is applied to the training dataset of each model, such that the models are trained to half the volume of data. The description of experiment 2b has been updated as follows (L249-250):

> "Next in experiment 2b, experiment 2a is repeated, but with cluster-based streamflow undersampling applied to the training dataset. The resulting models are trained to half the amount of training data as those in 2a."

R2-9: Line 248: Can you provide the names/locations of the basins/gages for those unfamiliar with the labels of the basins?

Thank you for this recommendation. We have added the names and provinces to the first mentions of each basin code, and a reference to the HYDAT database where the codes can be queried. For example, the updated figure 1 caption is as follows (L213):

> "Left column, from top to bottom: unclustered (a), clustered (c), and undersampled streamflow (e) for basin 01AD003 (HYDAT ID; located along St. Francis River in New Brunswick)."

R2-10: Figure 2: How does CUS-Q work for multiple features, such as streamflow, gradient, and the two day of year features? I.e., how do you define the distance between samples with multiple features in K-means?

The CUS-Q feature set includes all of the features described. Since CUS-Q is applied to basins individually, this results in a matrix with a shape of 4380 (samples) by 5 clustering features). The standard Euclidian distance metric is used in K-means. The description of the CUS-Q method has been rewritten (L184-189):

> "The engineered feature set includes streamflow ($q_t$), two streamflow gradient features (given as ($q_{t-3} - q_t$)/3 and $q_{t-1} - q_t$), and two day of year features (given as sin ($2\pi t/365$) and cos ($2\pi t/365$) where t is the day of year {1,...,365}. While the day of year is discontinuous between 365 and 1, the sin and cos decomposition offers a continuously changing pairing across each year, which increases the likelihood of clusters spanning from December to January. The streamflow gradient encourages the representation of rising and falling limbs within the clusters, which are not distinguishable using only the streamflow state."

R2-11: Figure 4: It is hard to see the solid black line in Figure 4.

We apologise for the lack of clarity in the figure and have adjusted the line colours and styles in the revised manuscript. The updated figure is included as Figure RC2b below.

[Figure]

[Figure]

*Figure RC2b: revised Figure 4 with reduced linewidths.*

R2-12: Line 254-255: Please rephrase this sentence.

We apologise for the unclear wording and have rephrased the sentence as follows (L267-269):

> "The performance of models trained on a set of 64 randomly sampled basins is shown in Fig. 4 in terms of NSE (a-c) and PI (d-f) for three cases: without resampling, with cluster-based temporal undersampling CUS_Q, and random temporal undersampling."

R2-13: Line 277: What are the testing basins for all configurations? Are they all 128 basins? It seems the undersampling of basins would harm model performance to some extent, and the benefits of CUS_B are limited.

Correct, models are evaluated on all 128 basins. The purpose of this experiment is not to achieve better performance than the baseline model, it is to determine the extent to which models trained to a subset of basins (identified using spatial clustering) can generalise to the entire set.

R2-14: Line 306-307: This finding is really interesting. I wonder if this is because the cluster number (2) is too small, causing them to still share many similarities. It is worth studying with a larger number of clusters.

We agree that a larger number of clusters would be better, however, larger numbers of clusters result in clusters that do not contain enough basins for validation. A workaround would be to

specify a minimum cluster constraint on the K-means clustering algorithm, however, that results in clusters with low cohesion, as quantified using the silhouette score. The relationship between the minimum cluster size average silhouette scores are included in Figure S1 in the Supplementary Information, and below in Figure RC2c.

[Figure]

**Figure S1.** Silhouette scores (left) for minimum cluster sizes of 2 (a), 8 (c), 16 (e), 32 (g), and 64 (i). Respective clustering results (right) for K values (corresponding to maximum silhouette score) of 8 (b), 7 (d, f), and 2 (h, j). World Gray Canvas basemap provided by ESRI.

*Figure RC2c: Relationship between silhouette score and minimum cluster size.*

R2-15: Line 317: I suggest moving all the discussion below to a separate discussion section.

Thank you for this recommendation - we have added a separate discussion subsection (Sec. 3.3.1) in the revised manuscript (L332).

R2-16: Figure 10: This figure is hard to read and understand.

We apologise for the lack of clarity. In our revised manuscript, we have split the subplots in half, so that each 'experiment' and metric has its own subplot. We have also added the experiment name to the subplot titles, with the hope that it improves the interpretability. See example in Figure RC2d.

[Figure]

Figure RC2d: Updated Figure 10 with an increased number of subplots and detailed subplot titles.

R2-17: Line 319-321: Which boxes in Figure 10 correspond to these two cases?

The result is true for both experiments. We have adjusted the text to reference the modified figure, in which experiments are contained in distinct subplots to improve clarity (L326-331):

"In experiment 2b the configurations from 2a are repeated, but incorporating CUSQ, which is introduced in Sec. 3.1. Models from 2a and 2b are compared in Fig. 10 for C0 and C1 evaluation sets. Consistent with the results from experiment 1a in Sec. 3.1, CUSQ is found to have very little impact on model test performance, which is illustrated by the similar results shown when comparing results without and with CUSQ undersampling, in subplot pairs (a)-(c), (b)-(d), (e)-(g), and (f)-(h). This result indicates that for fixed training dataset size, data from outside the region of interest is much more useful to the training procedure than redundant data within the region of interest, identifiable using the clustering procedure."

R2-18: Figure 10: The number of evaluation basins (4) is too small to represent the spatial generalizability of the model.

We agree that the sample size is a limitation of this experiment and have rerun the experiment to include another set of 4 basins. Since it's difficult to aggregate the figure included in the submitted manuscript, the new figures (RC2e and RC2f) are shown below, with the latter added to the the SI of the revised manuscript.

[Figure]

*Figure RC2e: Original experimental results.*

[Figure]

*Figure RC2f: Additional experimental run on two sets of 4 different basins*

**Additional changes**

Following feedback from a third-party reviewer, changes were made to experiments 1b and 2a, which previously contained cases where models were evaluated on basins that were not included among the one-hot encoded labels in the training dataset. In experiment 1b, the experiment was rerun without one-hot encoded labels in the training dataset. The following explanation is included in the revised manuscript (L255-257):

> "One distinction in this experimental configuration is that it does not use one-hot-encoded basin labels, as many of the test basins are not included in the training dataset. The labels are removed to ensure that the model does not develop any dependencies on them, since they are not available for the test basins."

In experiment 2a, 2 out of 5 configurations included test data that was not included in the training dataset. These were simply removed, as they were not deemed necessary to meet the objectives. The associated text in Section 3.1.1, that discussed transfer performance across basins, was also removed.

---

## Author Response (AR2)

**Reviewer 3 feedback**

I am satisfied with the authors' response and revision. I only have two small points for them to address. Other than that, I think this work is ready for publication.

Thank you for the taking the time to review our manuscript.

**General comments:**

Line 104: Please justify why the AR model, rather than the LSTM without AR, is used in this work. Also, (if possible,) add some discussion about the potential impacts on the conclusions if the non-AR LSTM were used.

We've added the justification for an AR-model on L104 and a brief comment on the possible effects:

L105: AR LSTMs are used since they are more accurate than non-AR LSTMs (Nearing, et al., 2022), and because the Canadian hydrometric network is largely available in real-time.

L288: A limitation of this experiment, and the subsequent experiments, is that they were conducted using AR LSTMs. If AR inputs are not available, the model hyperparameters would need to be reconfigured and model performance would be expected to decrease. While the experimental results are expected to transfer to non-AR models, this would need to be empirically confirmed.

Line 255-257: Based on my understanding and the response from the authors, experiment 2b repeats 2a with cluster-based temporal undersampling. Here, it should be 'This experiment provides a comparison point between models trained in 32 basins without CUS_Q and models trained on 64 basins with CUS_Q', not CUS_B. Please confirm it.

Thank you for identifying this error; the section has been revised:

L252: This experiment provides a comparison point between models trained in 32 basins without $CUS_Q$ and models trained on 64 basins with $CUS_Q$, as both configurations have the same number of samples.